# 3D genomic features across >50 diverse cell types reveal insights into the genomic architecture of childhood obesity

Khanh B Trang[1,2], Matthew C Pahl[1,2], James A Pippin[1,2], Chun Su[1,2], Sheridan H Littleton[1,2,3,4], Prabhat Sharma[1,5], Nikhil N Kulkarni[1,5], Louis R Ghanem[6], Natalie A Terry[6], Joan M O'Brien[7,8], Yadav Wagley[9], Kurt D Hankenson[9], Ashley Jermusyk[10], Jason Hoskins[10], Laufey T Amundadottir[10], Mai Xu[10], Kevin Brown[10], Stewart Anderson[11,12], Wenli Yang[13,14], Paul Titchenell[13,15], Patrick Seale[13,14], Klaus H Kaestner[4,13], Laura Cook[16,17,18], Megan Levings[19,20,21], Babette S Zemel[6,22], Alessandra Chesi[1,23], Andrew D Wells[1,5,23,24], Struan FA Grant[1,2,4,13,22,25,26]*

[1]Center for Spatial and Functional Genomics, The Children's Hospital of Philadelphia, Philadelphia, United States; [2]Division of Human Genetics, The Children's Hospital of Philadelphia, Philadelphia, United States; [3]Cell and Molecular Biology Graduate Group, Perelman School of Medicine, University of Pennsylvania, Philadelphia, United States; [4]Department of Genetics, Perelman School of Medicine, University of Pennsylvania, Philadelphia, United States; [5]Department of Pathology, The Children's Hospital of Philadelphia, Philadelphia, United States; [6]Division of Gastroenterology, Hepatology, and Nutrition, Children's Hospital of Philadelphia, Philadelphia, United States; [7]Scheie Eye Institute, Department of Ophthalmology, Perelman School of Medicine, University of Pennsylvania, Philadelphia, United States; [8]Penn Medicine Center for Ophthalmic Genetics in Complex Disease, Philadelphia, United States; [9]Department of Orthopedic Surgery University of Michigan Medical School Ann Arbor, Ann Arbor, United States; [10]Laboratory of Translational Genomics, Division of Cancer Epidemiology and Genetics, National Cancer Institute, Bethesda, United States; [11]Department of Child and Adolescent Psychiatry, Children's Hospital of Philadelphia, Philadelphia, United States; [12]Department of Psychiatry, Perelman School of Medicine, University of Pennsylvania, Philadelphia, United States; [13]Institute for Diabetes, Obesity and Metabolism, Perelman School of Medicine, University of Pennsylvania, Philadelphia, United States; [14]Department of Cell and Developmental Biology, Perelman School of Medicine, University of Pennsylvania, Philadelphia, United States; [15]Department of Physiology, Perelman School of Medicine, University of Pennsylvania, Philadelphia, United States; [16]Department of Microbiology and Immunology, University of Melbourne, Peter Doherty Institute for Infection and Immunity, Melbourne, Australia; [17]Department of Critical Care, Melbourne Medical School, University of Melbourne, Melbourne, Australia; [18]Division of Infectious Diseases, Department of Medicine, University of British Columbia, Vancouver, Canada; [19]Department of Surgery, University of British Columbia, Vancouver, Canada; [20]BC Children's Hospital Research Institute, Vancouver, Canada; [21]School of Biomedical Engineering, University of British Columbia, Vancouver, Canada; [22]Department of Pediatrics, Perelman School of Medicine, University of Pennsylvania,

*For correspondence:
grants@chop.edu

Competing interest: The authors declare that no competing interests exist.

Philadelphia, United States; [23]Department of Pathology, Perelman School of Medicine, University of Pennsylvania, Philadelphia, United States; [24]Institute for Immunology, Perelman School of Medicine, University of Pennsylvania, Philadelphia, United States; [25]Division Endocrinology and Diabetes, The Children's Hospital of Philadelphia, Philadelphia, United States; [26]Penn Neurodegeneration Genomics Center, Perelman School of Medicine, University of Pennsylvania, Philadelphia, United States

## eLife Assessment

This **important** study presents genome-wide high-resolution chromatin-based 3D genomic interaction maps for over 50 diverse human cell types and integrates these data with pediatric obesity GWAS. The work provides **convincing** evidence that multiple pancreatic islet cell types are key effector cell types. The authors also perform variant-to-gene mapping to nominate genes underlying several GWAS hits. Overall, the results will be of interest to both the fields of 3D genome architecture and pediatric obesity.

**Abstract** The prevalence of childhood obesity is increasing worldwide, along with the associated common comorbidities of type 2 diabetes and cardiovascular disease in later life. Motivated by evidence for a strong genetic component, our prior genome-wide association study (GWAS) efforts for childhood obesity revealed 19 independent signals for the trait; however, the mechanism of action of these loci remains to be elucidated. To molecularly characterize these childhood obesity loci, we sought to determine the underlying causal variants and the corresponding effector genes within diverse cellular contexts. Integrating childhood obesity GWAS summary statistics with our existing 3D genomic datasets for 57 human cell types, consisting of high-resolution promoter-focused Capture-C/Hi-C, ATAC-seq, and RNA-seq, we applied stratified LD score regression and calculated the proportion of genome-wide SNP heritability attributable to cell type-specific features, revealing pancreatic alpha cell enrichment as the most statistically significant. Subsequent chromatin contact-based fine-mapping was carried out for genome-wide significant childhood obesity loci and their linkage disequilibrium proxies to implicate effector genes, yielded the most abundant number of candidate variants and target genes at the *BDNF*, *ADCY3*, *TMEM18*, and *FTO* loci in skeletal muscle myotubes and the pancreatic beta-cell line, EndoC-BH1. One novel implicated effector gene, *ALKAL2* – an inflammation-responsive gene in nerve nociceptors – was observed at the key *TMEM18* locus across multiple immune cell types. Interestingly, this observation was also supported through colocalization analysis using expression quantitative trait loci (eQTL) derived from the Genotype-Tissue Expression (GTEx) dataset, supporting an inflammatory and neurologic component to the pathogenesis of childhood obesity. Our comprehensive appraisal of 3D genomic datasets generated in a myriad of different cell types provides genomic insights into pediatric obesity pathogenesis.

## Introduction

The prevalence of obesity has risen significantly worldwide (*NCD Risk Factor Collaboration (NCD-RisC), 2017*), especially among children and adolescents (*Bryan et al., 2021*). Obesity is associated with chronic diseases, such as diabetes, cardiovascular diseases, and certain cancers (*GBD 2015 Obesity Collaborators, 2017*; *Singh et al., 2013*; *Wormser et al., 2011*; *Lauby-Secretan et al., 2016*), along with mechanical issues including osteoarthritis and sleep apnea (*Fontaine and Barofsky, 2001*).

Modern lifestyle factors, including physical inactivity, excessive caloric intake, and socioeconomic inequity, along with disrupted sleep and microbiome, represent environmental risk factors for obesity pathogenesis. However, genetics also play a significant role, with the estimated heritability ranging from 40% to 70% (*Loos and Yeo, 2022*; *Maes et al., 1997*; *Elks et al., 2012*). Studies show that body weight and obesity remain stable from infancy to adulthood (*Demerath et al., 2007*; *Dubois*

*et al., 2007*; *Wardle et al., 2008*; *Bouchard, 2009*), but variation between individuals does exist (*Littleton et al., 2020*). Genome-wide association studies (GWAS) have improved our understanding of the genetic contribution to childhood obesity (*Vogelezang et al., 2020*; *Yaghootkar et al., 2020*; *Couto Alves et al., 2019*; *Fu et al., 2019*; *Turcot et al., 2018*; *Littleton and Grant, 2022*). However, the functional consequences and molecular mechanisms of identified genetic variants in such GWAS efforts are yet to be fully elucidated. Efforts are now being made to predict target effector genes and explore potential drug targets using various computational and experimental approaches (*Yu et al., 2022*; *Avsec et al., 2021*; *Gazal et al., 2022*; *Zhou and Troyanskaya, 2015*; *Nasser et al., 2021*), which subsequently warrant functional follow-up efforts.

With our extensive datasets generated on a range of different cell types, by combining 3D chromatin maps (Hi-C, Capture-C) with matched transcriptome (RNA-seq) and chromatin accessibility data (ATAC-seq), we investigated heritability patterns of pediatric obesity-associated variants and their gene-regulatory functions in a cell type-specific manner. This approach yielded 94 candidate causal variants mapped to their putative effector gene(s) and corresponding cell type(s) setting. In addition, using methods comparable to our prior efforts in other disease contexts (*Chesi et al., 2019*; *Su et al., 2020*; *Pahl et al., 2021*; *Cousminer et al., 2021*; *Vujkovic et al., 2022*; *Pahl et al., 2022*; *Su et al., 2022*; *Palermo et al., 2023*), we also uncovered new variant-to-gene combinations within specific novel cellular settings, most notably in immune cell types, which further confirmed the involvement of the immune system in the pathogenesis of obesity in the early stages of life.

## Results

### Enrichment assessment of childhood obesity variants across cell types

To explore the enrichment of childhood obesity GWAS variants across cell types, we carried out Partitioned Linkage Disequilibrium Score Regression (LDSR) (*Finucane et al., 2015*) on all ATAC-seq-defined OCRs for each cell type. We assessed cell-type-specific enrichment of GWAS signals in four main categories of genomic regions (*Figure 1A*): (1) Total OCRs: open chromatin regions defined by ATAC-seq; (2) Promoter OCRs: the subset of OCRs overlapping a gene promoter; (3) cREs: the subset of OCRs that form chromatin loops (as determined by Hi-C/Promoter Capture-C) with a gene promoter, and are therefore considered putative enhancers or suppressors regulating gene expression; (4) cREs ±500bases: extended cREs by 500 bases in both directions. The rationale behind this approach is that different GWAS variants can influence phenotypes by regulating gene expression in a cell-type specific manner through various regulatory mechanisms. For example, they may alter enhancer function (cREs category) or affect the binding of a transcription factor at a gene's promoter (Promoter OCRs category).

We observed that 41 of 57 cell types – including 22 metabolic, 21 immune, 7 neural cell types, and 7 independent cell lines (*Supplementary file 1a*) – showed at least a degree of directional enrichment with the total set of OCRs (*Figure 1B* – Total OCRs). However, only four cell types – two pancreatic alpha and two pancreatic beta cell-based datasets – had statistically significant enrichments (p<0.05). These enrichments were less pronounced when focusing on promoter OCRs only (*Figure 1B* – Promoter OCRs). To further limit the LD enrichment assessment to just those OCRs that can putatively regulate gene expression via chromatin contacts with gene promoters, we used the putative cREs (*Chesi et al., 2019*; *Pahl et al., 2021*). This reduced the number of cell types showing at least nominal enrichment (31 of 57), enlarged the dispersion of enrichment ranges across different cell types, increased the 95% confidence intervals (CI) of enrichments, and hence increased the *P*-value of the resulting regression score. cREs from pancreatic alpha cells derived from single-cell ATAC-seq were the only dataset that remained statically significant (*Figure 1B* – putative cREs).

The original reported LDSR method analyzed enrichment in the 500 bp flanking regions of their regulatory categories (*Finucane et al., 2015*). However, when we expanded our analysis to the ± 500 bp window for our cREs, albeit incorporating more weighted variants into the enrichment (represented by larger dots in *Figure 1B* – cREs ± 500 bases), this resulted in a decrease in the number of cell types yielding at least nominal enrichment (26 cell types), the enrichment range across cell types, the 95% CI, and level of significance. The pancreatic alpha cell observation also dropped below the bar for significance with this expanded window definition.

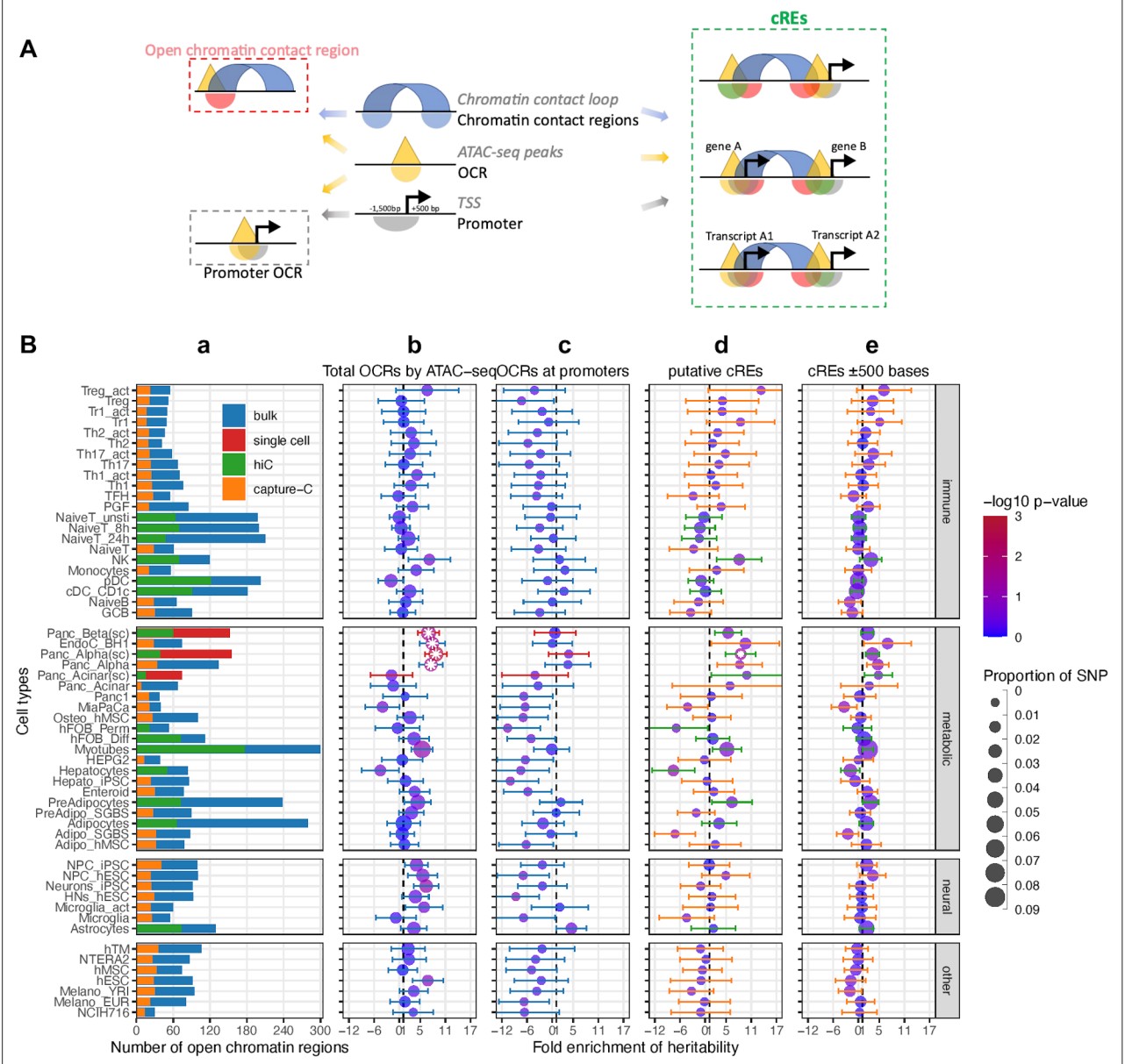

**Figure 1.** Partitioned Linkage Disequilibrium Score Regression analysis for open chromatin regions of all cell types. (**A**) The schematic shows the different types of regions defined in our study and 3 different ways overlapping chromatin contact regions – OCRs – gene promoters define cREs. (**B**) Heritability enrichment by LDSC analysis for each cell type. (**a**) Bar-plot shows the total number of OCRs identified by ATAC-seq for each cell type on bulk cells - blue, or on single cells – red; and the portion of OCRs that fall within cREs identified by incorporating Hi-C (green) or by Capture-C (orange). 4 panels of dot-plots show heritability enrichment by LDSC analysis for each cell type, with standard error whiskers. Dots' colors correspond to -log10(p-values), dots with white asterisks are significant *P*-values <0.05, and dots' sizes corresponding to the proportion of SNP contribute to heritability. Dash line at 1, i.e., no enrichmen. (**b**) Analysis done on whole OCRs set of each cell type (whiskers colors match with bulk/single cell from bar-plot a); (**c**) On only OCRs that overlapped with promoters (whiskers' colors match with 623 bulk/single cell from bar-plot a); (**d**) On the putative cREs of each cell type (whiskers' colors match with Hi-C/Capture-C from bar-plot a); (**e**) On the same cREs as (**c**) panel with their genomic positions expanded ±500 bases on both sides (whiskers' colors match with Hi-C/Capture-C from bar-plot a).

## Consistency and diversity of childhood obesity proxy variants mapped to cREs

Despite the enrichments above only being limited to just a small number of cell types, it is likely that individual loci have differing levels of contributions in various cellular contexts and could not be detected at the genome wide assessment scale. As such we elected to further explore the candidate

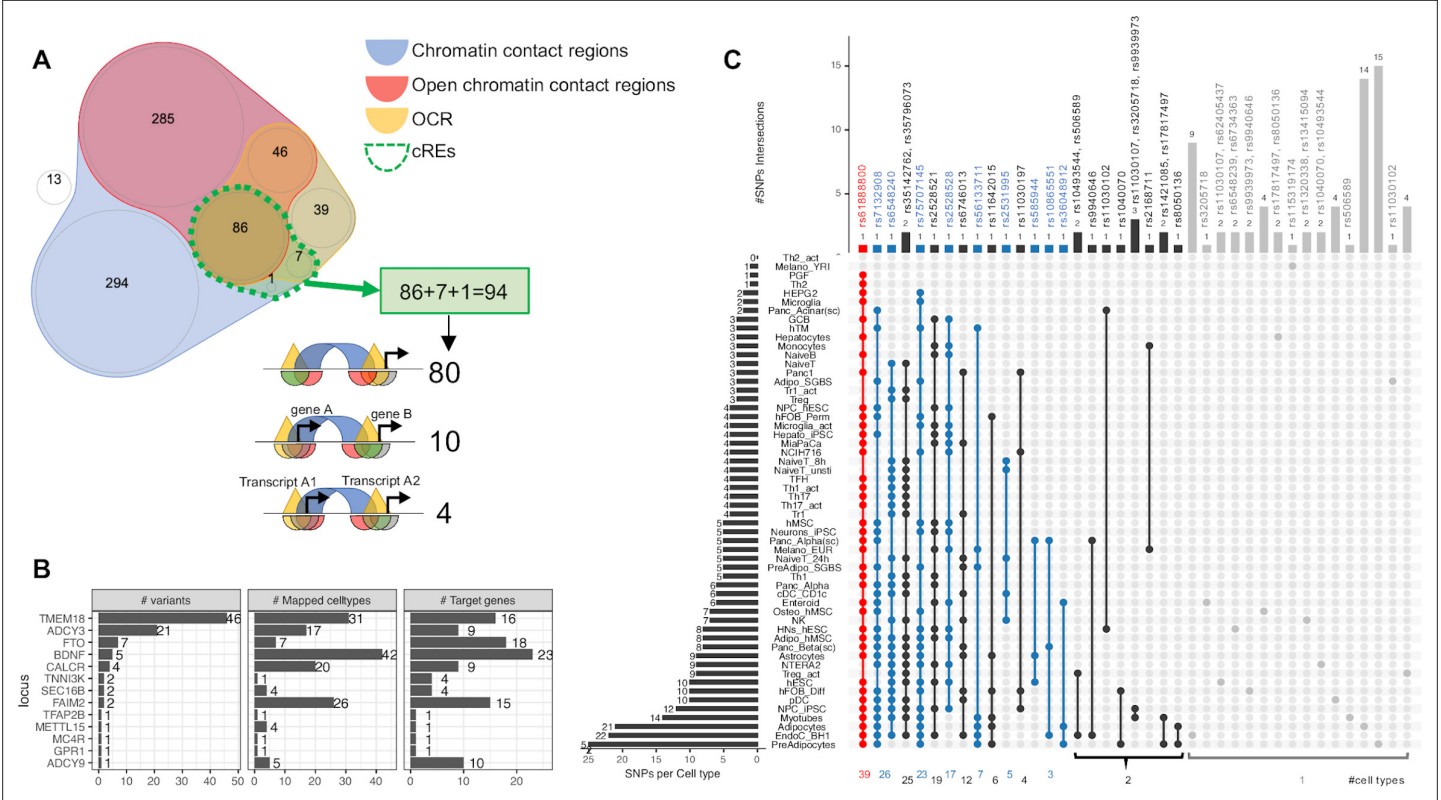

**Figure 2.** Mapping 771 proxies to the open chromatin regions of each cell type. (**A**) Venn diagram shows how 771 proxies mapped to the OCRs: ·
Blue area: 758 proxies were located within contact regions of at least one cell type regardless of chromatin state; · Red area: 417 proxies were located
within contact regions marked as open by overlapping with OCR; · Yellow area: If we only considered open chromatin regions, 178 proxies were
included; · Dotted green bordered area: To focus on just those variants residing within open chromatin and contacting promoter regions in any cell
type, we overlapped the genomic positions of these proxies with each cell type's cRE set, yielding 90 variants (3 from the 99% credible set) directly
contacting open gene promoters (*Supplementary file 1b*), with 10 of which located within a promoter of one gene but contacting another different
gene promoter. There were an additional 4 variants located within gene promoters but in chromatin contact with promoter(s) of nearby transcript(s) of
the same gene (correspond to 3 cREs illustrations in *Figure 1A*). · White area: proxies that fall into neither defined region of interest. (**B**) Bar-plot shows
number of proxies, cell types and target genes mapped at each locus. (**C**) The upSet plot shows the degree of overlap across cell types of the variants;
ranked from the most common variant (red) – rs61888800 from *BDNF* locus, a well-known 5' untranslated region variant of this gene that is associated
with anti-depression and therapeutic response (*Licinio et al., 2009*; *Colle et al., 2015*) – appeared in 39 cell types, to the group of variants (grey) which
appeared in only one cell type.

The online version of this article includes the following figure supplement(s) for figure 2:

**Figure supplement 1.** Venn diagrams show intersections and the number of proxies within each locus were mapped in different scenarios as illustrated
in *Figure 1A*.

**Figure supplement 2.** The upSet plot outlines in which cell type(s) each locus appeared.

**Figure supplement 3.** Examples of 2 variants mapped to only one gene promoter.

**Figure supplement 4.** Dot-plot with each variant colored by their locus shows the number of genes and cell types each proxy mapped into our cREs.

**Figure supplement 5.** Bar plots for the number of cell types and number of genes implicated by each variant.

**Figure supplement 6.** TMEM18 locus: within this locus where there were 45 proxies in LD with the sentinel SNP rs7579427 that mapped to the cREs of
31 cell types; 35 of these proxies were found exclusively in only one unique cell type (10 in pre-adipocytes only, 9 in adipocytes only, 8 in EndoC-BH1
only, 2 in hESC-derived hypothalamic neurons only, 2 in natural killer cells only, and 4 in activated regulatory T cells only); however, their target gene
promoters were also frequently contacted by other different proxies in other cell types.

effector genes that are directly affected by cREs harboring childhood obesity-associated variants by
systematically mapping the genomic positions of the LD proxies onto each cell type's cREs. Most
proxies fall within chromatin contact regions (blue area in Venn diagram *Figure 2A*) or OCRs (yellow
area) or open chromatin contact regions (red area), or completely outside (white area) any defined
region. Only 94 proxies fall within our defined cREs (overlapped area with dotted green border in

*Figure 2A*), they clustered at 13 original loci (*Supplementary file 1b*). *Figure 2—figure supplement 1* outlines the number of signals at each locus included or excluded based on the criteria we defined for our regions of interest. The *TMEM18* locus yielded the most variants through cREs mapping, with 46 proxies for the two lead independent variants, rs7579427 and rs62104180. The second most abundant locus was *ADCY3*, with 21 proxies for lead variant rs4077678 (*Figure 2B*). The higher number of variants at one locus did not correlate with implicating more genes or cell types through mapping. The mapping frequency of various variants within a specific locus exhibited substantial differences (*Figure 2—figure supplement 2*).

Inspecting individual variants regardless of their locus, we found that 28 of 94 proxies appeared in cREs across multiple cell types, with another 66 observed in just one cell type (*Figure 2C*). 45 variants of these 66 just contacted one gene promoter, such as at the *GPR1* and *TFAP2B* loci (*Figure 2—figure supplement 3*).

Overall, the number of cell types in which a variant was observed in open chromatin correlated with the number of genes contacted via chromatin loops (*Figure 2—figure supplement 4*). However, we also observed that some variants found in cREs in multiple cell types were more selective with respect to their candidate effector genes (*Figure 2—figure supplement 5* - red arrow), or conversely, more selective across given cell types but implicated multiple genes (*Figure 2—figure supplement 5* - blue arrow). *Figure 2—figure supplement 6* outlines our observations at the *TMEM18* locus – an example locus involved in both scenarios.

## Implicated genes cluster at loci strongly associated with childhood obesity consistently across multiple cell types

Mapping the variants across all the cell types resulted in a total of 111 implicated childhood obesity candidate effector genes (*Table 1*). Among these, 45 genes were specific to just one cell type (*Figure 3—figure supplement 1*), including 13 in myotubes and 7 in natural killer cells. Conversely and notably, *BDNF* appeared across 42 different cell types. Across the metabolic, neural, and immune systems and seven other cell lines, there were nine genes consistently implicated in all four categories (top panel *Figure 3* – red stars, *Figure 3—figure supplement 2* 'all'), while five genes were consistently implicated in metabolic, neural, and immune systems (top panel *Figure 3* – blue stars, *Figure 3—figure supplement 2* 'all_main'). Two genes, *ADCY3* and *BDNF*, had variants both at their promoters and contacted variants in cREs via chromatin loops (*Figure 3—figure supplement 3*).

At the *TMEM18* locus on chr 2p25.3, a highly significant human obesity locus that has long been associated with both adult and childhood obesity, we observed differing degrees of evidence for 16 genes, but noted that rs6548240, rs35796073, and rs35142762 consistently contacted the *SH3YL1*, *ACP1*, and *ALKAL2* promoters across multiple cell types (*Figure 2C* -third and fourth column).

At the chr 2p23 locus, *ADCY3* yielded the most contacts (i.e. many proxies contacting the same gene via chromatin loops), suggesting this locus acts as a regulatory hub. However, we observed a similar composition in cell types for four other genes: *DNAJC27*, *DNAJC27-AS1* (both previously implicated in obesity and/or diabetes traits *Cherian et al., 2018*), *AC013267.1*, and *SNORD14* (*RF00016*). *ITSN2*, *NCOA1*, and *EFR3B* were three genes within this locus that were only implicated in immune cell types. *NCOA1* encodes a prominent meta-inflammation factor (*Rollins et al., 2015*) known to reduce adipogenesis and shift the energy balance between white and brown fat, and its absence known to induce obesity (*Mohsen G et al., 2019*).

*CALCR* was the most frequently implicated gene at its locus, supported by 20 cell types across all systems. While within the *BDNF* locus, *METTL15* and *KIF18A* – two non-cell-type-specific genes - plus some lncRNA genes, were contacted by childhood obesity-associated proxies within the same multiple cell types as *BNDF*, again suggesting the presence of a regulatory hub.

At the *FAIM2* locus on chr 12q13.12, we observed known genes associated with obesity, eating patterns, and diabetes-related traits, including *ASIC1*, *AQP2*, *AQP5*, *AQP6*, *RACGAP1*, and *AC025154.2 (AQP5-AS1)* along with *FAIM2* (*Table 1*). These genes were harbored within cREs of astrocytes, neural progenitors, hypothalamic neurons, and multiple metabolic cell types. Plasmacytoid and CD1c+conventional dendritic cells were the only two immune cell types that harbored such proxies within their cREs, implicating *ASIC1*, *PRPF40B*, *RPL35AP28*, *TMBIM6*, and *LSM6P2* at the *FAIM2* locus.

**Table 1.** PubMed-query known functions for 111 genes implicated by obesity variants.

| Locus | Implicated genes | Obesity or related traits | Different traits |
|---|---|---|---|
| TNNI3K | LRRIQ3 | (NA) | Associated with opioid usage [PMID:34728798] and MDD [PMID: 31748543] |
| | FPGT | Predict BMI in Korean pop. [PMID: 28674662] | (NA) |
| | FPGT-TNNI3K | | Associated with MDD [PMID: 31748543] |
| | LRRC53 | Associated with high BMI increased risk heart attack [PMID: 32471361] | (NA) |
| | ASTN1 | Identified as obesity QTL in rat [PMID: 35729251] | Associated with neurodevelopmental traits [PMID: 24381304] and variety of cancers [PMID: 32945491] |
| | BRINP2 | (NA) | Associated with neurodevelopmental traits [PMID: 34267256] |
| SEC16B | AL122019.1 | (NA) | |
| | AL162431.1 | | |
| TMEM18 | FAM110C | (NA) | Overexpression induces microtubule aberrancies [PMID: 17499476], involved in cell spreading and migration [PMID: 19698782] |
| | SH3YL1 | Associated with BMI in type 2 diabetes nephropathy [PMID: 33223406] | Influence on T cell activation [PMID: 31427643], involved in different cancer types [PMID: 26305679,24508479] |
| | ACP1 | Associated with early-onset obesity [PMID: 24129437], correlated with cardiovascular risks [PMID: 19570551], drive adipocyte differentiation via control of pdgfrα signaling [PMID: 33615467] | Associated with bipolar disorder [PMID: 31830721] |
| | ALKAL2 | Associated with childhood BMI [PMID: 33627773] | Enhance expression in response to inflammatory pain in nociceptors [PMID: 35608912, 35610945] |
| | MYT1L | Associated with early-onset obesity [PMID: 24129437] | (NA) |
| | AC079779.1 | (NA) | |
| | AC079779.2 | | |
| | AC079779.3 | | |
| | AC079779.4 | | |
| | LINC01865 | | |
| | AC105393.2 | | |
| | AC105393.1 | | |
| | LINC01874 | | |
| | LINC01875 | | |
| | AC093326.1 | | |
| | AC141930.2 | | |

*Table 1 continued on next page*

*Table 1 continued*

| Locus | Implicated genes | Obesity or related traits | Different traits |
|---|---|---|---|
|  | ITSN2 | (NA) | Regulate T-cells function [PMID: 32618424] and help the interaction with B-cells [PMID: 29337666] |
|  | NCOA1 | Meta-inflammation gene [PMID: 25647480], reduce adipogenesis, shift the energy balance between white and brown fat [PMID: 31133421] |  |
|  | ADCY3 | Regulate/impair MC4R within energy-regulating melanocortin signaling pathway [PMID: 29311635,32955435] |  |
| ADCY3 | DNAJC27-AS1 | Linked to obesity, diabetes traits [PMID: 30131766] | (NA) |
|  | DNAJC27 | Linked to obesity, diabetes traits [PMID: 30131766] |  |
|  | EFR3B | Associated with T1D [PMID: 21980299], down-regulated in rare obesity-related disorder [PMID: 25705109] |  |
|  | WDR43 | (NA) | Associated with breast cancer [PMID: 27117709] |
|  | AC013267.1 | (NA) |  |
|  | RF00016 |  |  |
| GPR1 | GPR1 | Increase expression in obese phenotype [PMID: 34174278] | (NA) |
| TFAP2B | TFAP2D |  | Involve in embryogenesis [PMID: 12711551] |
|  | HEPACAM2 |  | Associated with colorectal cancer [PMID: 29659199, 29973580] |
|  | VPS50 | (NA) | Involve in neurodevelopmental disorders and defects [PMID: 30828385, 34037727] |
|  | MIR653 |  | Involve in different types of cancer [PMID: 35777307] |
|  | MIR489 | Promote adipogenesis in mice [PMID: 34004251] |  |
| CALCR | CALCR | Associated with BMI and control of food-intake [PMID: 34462445, 34210852, 31955990, 29522093] | (NA) |
|  | TFPI2 | (NA) | Involved in colorectal cancer [PMID: 35004840, 34092617, 25902909] |
|  | BET1 | Involved in triacylglycerol metabolism [PMID: 24423365] | Associated with muscular dystrophy [PMID: 34310943, 34779586] |
|  | AC003092.1 | (NA) | Association with glioblastoma [PMID: 33815468, 30442884] |
|  | AC002076.1 | (NA) |  |

*Table 1 continued on next page*

*Table 1 continued*

| Locus | Implicated genes | Obesity or related traits | Different traits |
|---|---|---|---|
| | LIN7C | Associated in T2D [PMID: 20215397], obesity [PMID: 23044507] | Associated with psychopathology [PMID: 23044507] |
| | BDNF-AS | Regulate *BDNF* and *LIN7C* expression [PMID: 22960213, 22446693] | (NA) |
| | BDNF | Regulate eating behavior and energy balance [PMID: 34556834] | |
| | MIR610 | (NA) | Involve in different types of cancer [PMID: 34408418, 29228616, 26885452] |
| | KIF18A | | Involve in different types of cancer [PMID: 35591854, 35286090] |
| | METTL15 | Associated with childhood obesity [PMID: 31504550] | (NA) |
| | AC090124.1 | (NA) | Reported to differentially prognostic of pancreatic cancer [PMID: 34307375] |
| | ARL14EP | | Involve in WAGR syndrome [PMID: 36011342, 31511512] |
| | DCDC1 | | Involvement with eyes anomalies [PMID: 34773354, 34703991] |
| BDNF | THEM7P | | Associated with mechanisms underlying inguinal hernia [PMID: 34392144] |
| | AL035078.2 | | |
| | ELP4 | | |
| | LINC00678 | | |
| | AC023206.1 | | |
| | RN7SKP158 | | |
| | AC104978.1 | | |
| | MIR8068 | (NA) | |
| | AC013714.1 | | |
| | AC100773.1 | | |
| | AC090833.1 | | |
| | AC090791.1 | | |
| | AC110056.1 | | |
| | AL035078.2 | | |

*Table 1 continued on next page*

*Table 1 continued*

| Locus | Implicated genes | Obesity or related traits | Different traits |
|---|---|---|---|
| | PRPF40B | (NA) | Splicing regulator involved in T-cell development [PMID: 31088860, 34323272] |
| | TMBIM6 | Deficiency leads to obesity by increasing Ca2+-dependent insulin secretion [PMID: 32394396] | Immune cell function and survival [PMID: 26470731] |
| | BCDIN3D | Associated with obesity, T2D [PMID: 20215397] | |
| | FAIM2 | Associated with childhood obesity [PMID: 31504550] | (NA) |
| | AQP2 | Associated with obesity, diabetes [PMID: 33367818] | |
| | AQP5 | Associated with non-obese diabetes [PMID: 25635992,22320885] | Responsible for transporting water, involve in Sjogren's syndrome [PMID: 25635992, 31557796] |
| FAIM2 | AQP6 | Down-regulated in retina in diabetes [PMID: 21851171] | Associated with renal diseases [PMID: 30654539] |
| | RACGAP1 | Involve in diabetes nephropathy [PMID: 35222021] | |
| | ASIC1 | Inhibition increase food intake and decrease energy expenditure [PMID: 35894166] | (NA) |
| | LSM6P2 | | |
| | RPL35AP28 | | |
| | LINC02396 | (NA) | |
| | LINC02395 | | |
| | AC025154.1 | | |
| | AC025154.2 | | |
| | SLX4 | (NA) | Associated with blood pressure [PMID: 30671673] |
| | DNASE1 | Associated with obesity hypertension [PMID: 33351325] | |
| | TRAP1 | Involve in global metabolic network, deletion reduce obesity incidence [PMID: 25088416] | (NA) |
| | CREBBP | Associated with high adiposity and low cardiometabolic risk [PMID: 33619380] | |
| ADCY9 | ADCY9 | Asoociated with BMI, obesity [PMID: 33619380, 23563607] | |
| | SRL | (NA) | Involve in cardiac dysfunction [PMID: 22119571] |
| | LINC01569 | | Associated with cancer and endometriosis [PMID: 35341703, 34422671] |
| | TFAP4 | Associated with BMI, birth weight, maternal glycemic [PMID: 35708509] | (NA) |
| | AC012676.1 | (NA) | Involve in hepatocellular carcinoma [PMID: 35210216] |
| | AC009171.2 | (NA) | |

*Table 1 continued*

| Locus | Implicated genes | Obesity or related traits | Different traits |
|---|---|---|---|
| FTO | FTO | Most extensively studied obesity locus [PMID: 34556834] | (NA) |
| | IRX3 | Obesogenic effects in adipocytes [PMID: 26760096], brain [PMID: 24646999], pancreas[93] | |
| | IRX5 | | |
| | AC018553.1 | (NA) | Associated with melanoma [PMID: 35611195] |
| | CRNDE | Regulator of angiogenesis in obesity-induced diabetes [PMID: 31863035] | (NA) |
| | MMP2 | Involve in obesity-relate angiogenesis [PMID: 35919566] | |
| | CAPNS2 | (NA) | Associated with thyroid-related traits [PMID: 23408906] |
| | AMFR | Involve in hepatic lipid metabolism [PMID: 33591966] | (NA) |
| | CETP | Involve in monogenic hyperalphalipoproteinemia [PMID: 34878751] | |
| | RPGRIP1L | Hypomorphism of this ciliary gene linked to morbid obesity [PMID: 27064284, 30597647, 29657248] | Required for hypothalamic arcuate neuron development [PMID: 30728336] |
| | LINC02169 | (NA) | Associated with occupational exposure to gases/fumes and mineral dust [PMID: 31152171] |
| | AC007491.1 | (NA) | |
| | AC018553.2 | | |
| | LINC02140 | | |
| | AC106738.1 | | |
| | AC106738.2 | | |
| | MTND5P34 | | |
| | AC007336.1 | | |
| MC4R | AC090771.1 | (NA) | |

The independent *ADCY9* and *FTO* loci are both located on chromosome 16. Genes at the *ADCY9* locus were only implicated in a subset of immune cell types. Interestingly, genes at the *FTO* locus were only implicated in Hi-C datasets (as opposed to Capture C), including 6 metabolic cell types and astrocytes. Most genes at the *FTO* locus were implicated in skeletal myotubes, differentiated osteoblasts, and astrocytes, namely *FTO* and *IRX3*; while *IRX5, CRNDE,* and *AC106738.1* were also implicated in adipocytes and hepatocytes.

## The most implicated cell types by two sets of analyses

EndoC-BH1 and myotubes are the two cell types in which we implicated the most effector genes, with 38 and 42, respectively – *Figure 3* side panel. This phenomenon is likely proportional in the case of myotubes, given the large number of cREs identified by overlapped Hi-C contact data and ATAC-seq open regions (*Figure 1A*), but not for EndoC-BH1. Albeit harboring an average number of cREs compared to other cell types, EndoC-BH1 cells were consistently among the top-ranked heritability estimates for the childhood obesity variants resulting from the EGG consortium GWAS (*Figure 1*) and harbored a significant number of implicated genes by the mapping of proxies. Interestingly, the pancreatic alpha cell type – shown above to be the most significant for heritability estimate by LDSC – revealed only six implicated genes contacted by the defined proxies, namely *BDNF* and five lncRNA genes.

## Pathway analysis

Of the 111 implicated genes in total, PubMed query revealed functional studies for 66 genes. The remaining were principally lncRNA and miRNA genes with currently undefined functions (*Table 1*). To

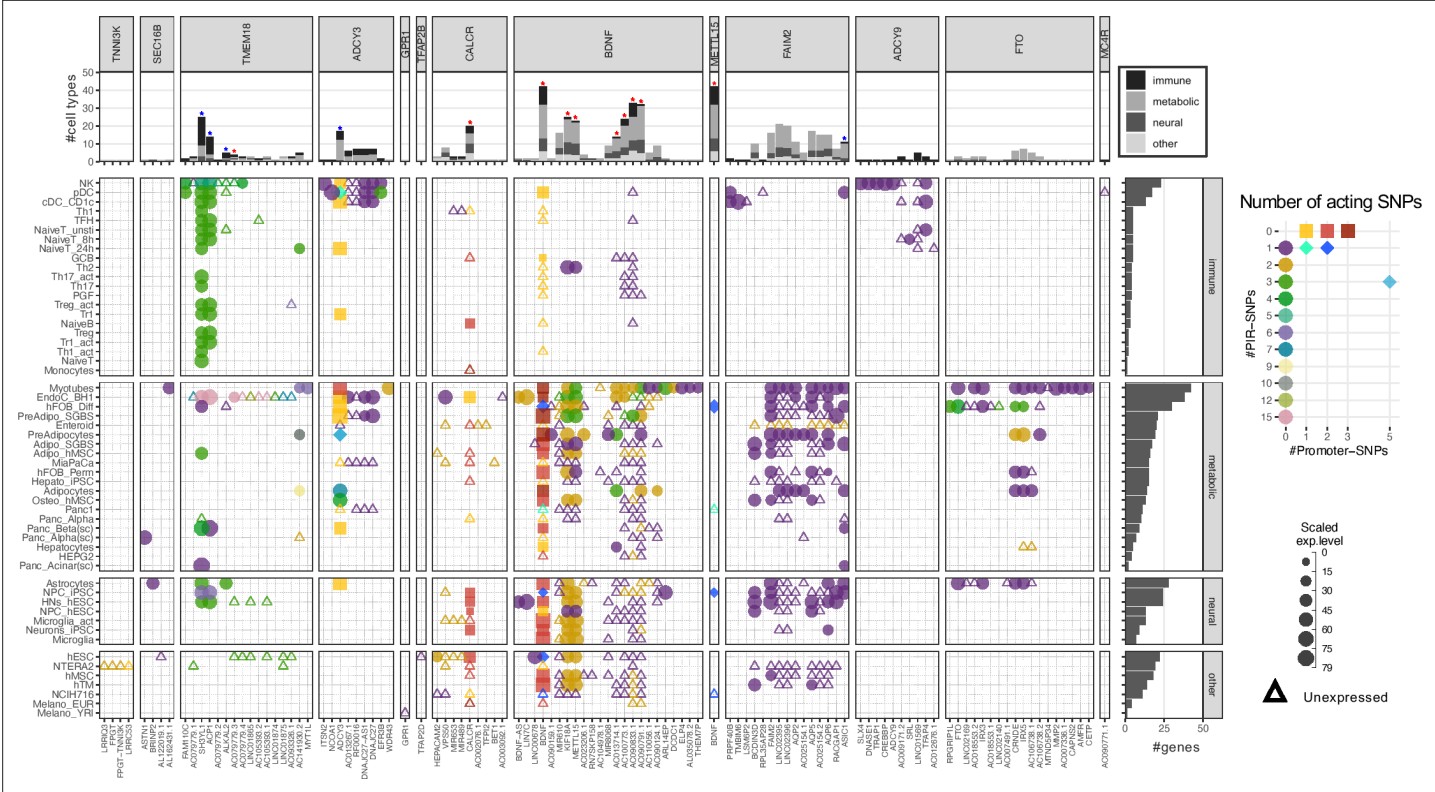

**Figure 3.** Profiles of 111 implicated genes by 94 proxies through cREs of each cell type. Main panel: Bubble plot show corresponding expression level (size) and number of variants (color) target each implicated gene of each cell type. Squares represent genes with variants at their promoters. Circles represent genes with variants contacted through chromatin loops. Some genes were implicated by both types, these 'double implications' are represented as diamond shapes, and were identified across several cell types: two cell types (plasmacytoid dendritic cells and pre-differentiated adipocytes) for *ADCY3* gene, and five for *BDNF* (human embryonic stem cells - hESC, differentiated human fetal osteoblast cells - hFOB_Diff, neural progenitor cells derived from induced pluripotent stem cells - NPC_iPSC, PANC-1, and NCIH716 cell lines). Genes with expression undetected in our arrays are shown as triangles. Top panel: bar-plot shows numbers of cell types each gene was implicated within, color-coded by which systems the cell types belong to. Right panel: bar-plot shows numbers of genes implicated by the variants with each cell type.

The online version of this article includes the following figure supplement(s) for figure 3:

**Figure supplement 1.** Bar-plot shows number of genes implicated in how many cell types.

**Figure supplement 2.** Bar-plot shows number of genes implicated in cell types of each combination of metabolic, immune, neural system and other cell lines groups.

**Figure supplement 3.** Double-implicated genes.

investigate how our implicated genes could confer obesity risk, we performed several pathway analyses keeping them either separated for each cell type or pooling into the respective metabolic, neural, or immune system sets. *Figure 4—figure supplement 1* shows simple Gene Ontology (GO) biological process terms enrichment results.

Leveraging the availability of our expression data generated via RNA-seq (available for 46 of 57 cell types), we performed pathway analysis. Given that our gene sets from the variant-to-gene process was stringently mapped, the sparse enrichment from normal direct analyses is not ideal for exploring obesity genetic etiology. Thus, we incorporated two methods from the *pathfindR* package (*Ulgen et al., 2019*) and our customized SPIA (details in Materials and methods, *Figure 4—figure supplement 2*). The result of 60 enriched KEGG terms is shown in *Figure 4A*, *Supplementary file 1c*, with 13 genes in 14 cell types for *pathfindR* and 39 enriched KEGG terms shown in *Figure 4*, *Supplementary file 1d*, with 10 genes in 42 cell types for customized SPIA. There were 20 overlapping pathways between the two approaches (yellow rows in *Supplementary file 1c-d*) including many signaling pathways such as the GnRH (hsa04912), cAMP (hsa04024), HIF-1 (hsa04066), Glucagon (hsa04922), Relaxin (hsa04926), Apelin (hsa04371), and Phospholipase D (hsa04072) signaling pathways. They

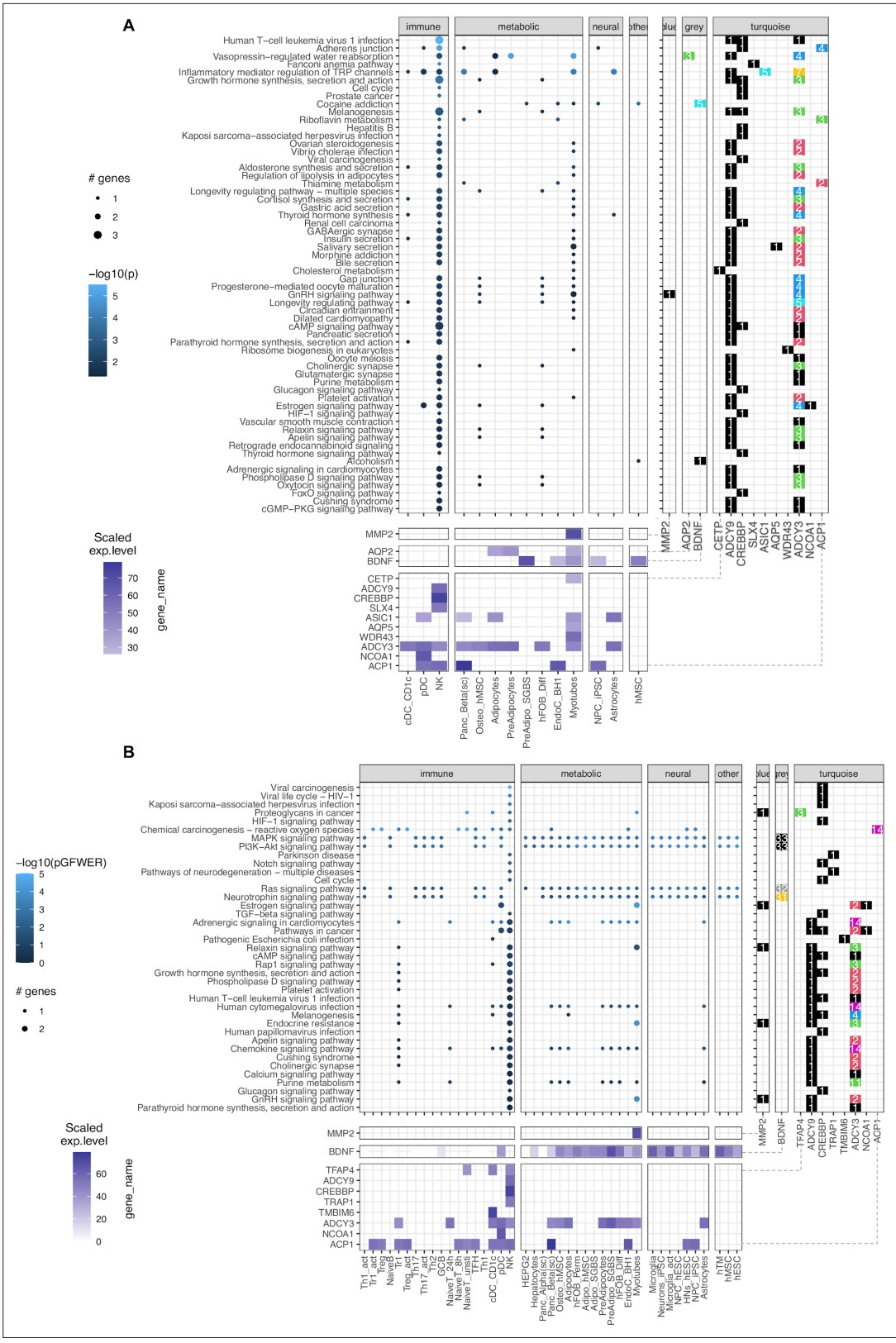

**Figure 4.** KEGG pathways enrichment analysis. (**A**) The pathfindR method: Focused on 'leveraging interaction information from a protein-protein interaction network (PIN) to identify distinct active subnetworks and then perform enrichment analyses on these subnetworks', thus aiding enriched pathway analyses through the inter-connection between the genes targeted by obesity variants with key genes driving the pathology of the disease.

*Figure 4 continued on next page*

*Figure 4 continued*

The result with 60 enriched KEGG terms in the main panel (***Supplementary file 1c***), shows 13 genes in 14 cell types and their scaled expression levels in the lower panel. (**B**) The modified SPIA: After applying the adjusted P-value of 0.05 as the filtering threshold, the analysis yielded 39 enriched KEGG terms (full table at ***Supplementary file 1d***) with only 10 genes, but involved up to 42 cell types.

The online version of this article includes the following figure supplement(s) for figure 4:

**Figure supplement 1.** Gene Ontology (GO) biological process terms enrichment.

**Figure supplement 2.** The additional metrics scheme.

**Figure supplement 3.** The GnRH signaling pathway: The KEGG graph shows the involvement of *ADCY3* and *MMP2* genes driving the GnRH signaling pathway.

**Figure supplement 4.** Cluster dendrogram of weighted genes expression of genes from the variant-to-genes mapping process into three modules, named by the colors.

were all driven by one or more of these 5 genes: *ADCY3*, *ADCY9*, *CREBBP*, *MMP2,* and *NCOA1*. Interestingly, we observed the involvement of natural killer cells in nearly all the enriched KEGG terms from *pathfindR* due to the high expression of the two adenylyl cyclase encoded genes, *ADCY3* and *ADCY9*, along with *CREBBP*. The SPIA approach disregarded the aquaporin genes (given they appear so frequently in so many pathways that involve cellular channels) but highlighted the central role of *BDNF* which single-handedly drove four signaling pathways: the Ras, Neurotrophin, PI3K-Akt, and MAPK signaling pathways. This also revealed the role of *TRAP1* in neurodegeneration.

These two approaches did not discount the role of *FAIM2* and *CALCR*. However, their absence was mainly due to the content of the current KEGG database. On the other hand, these approaches accentuated the role of the *MMP2* gene at the *FTO* locus in skeletal myotubes, given its consistency within the GnRH signaling pathway (***Figure 4—figure supplement 3***), which is in line with previous studies linking its expression with obesity (***Derosa et al., 2008***; ***Aksoyer Sezgin et al., 2022***; ***Nonino et al., 2021***).

## Supportive evidence by colocalization of target effector genes with eQTLs

The GTEx consortium has characterized thousands of eQTLs, albeit in heterogeneous bulk tissues (***Pejman, 2017***). To assess how many observed gene-SNP pairs agreed with our physical variant-to-gene mapping approach in our multiple separate cellular settings, we performed colocalization analysis using ColocQuiaL (***Chen et al., 2022***).

282 genes were reported to be associated with the variants within 13 loci from our variant-to-genes analysis. We found 114 colocalizations for ten of our loci that had high conditioned posterior probabilities (cond.PP.H4.abf≥0.8), involving 44 genes and 41 tissues among the eQTLs. We extracted the posterior probabilities for each SNP within each colocalization and selected the 95% credible set as the likely causal variants (complete list in ***Supplementary file 1e***). Despite sensitivity differences and varying cellular settings, when compared with our variant-to-gene mapping results, colocalization analysis yielded consistent identification for 21 pairs of SNP-gene interactions when considering the analyses across all our cell types, composed of 20 SNPs and 7 genes. Details of these SNP-gene pairs are shown in ***Figure 5A and B***.

Of these 20 SNPs, 15 were at the *ADCY3* locus, in LD with sentinel variant rs4077678, and all implicated *ADCY3* as the effector gene in 29 cell types – 15 metabolic, 6 immune, 4 neural cell types, and 4 independent cell lines (***Figure 5C***). Indeed, missense mutations have been previously reported for this gene in the context of obesity (***Grarup et al., 2018***; ***Stergiakouli et al., 2014***) while another member of this gene family, *ADCY5*, has also been extensively implicated in metabolic traits (***Sinnott-Armstrong et al., 2021***).

## Predicting transcription factors (TFs) binding disruption at implicated genes contributing to obesity risk

TFs regulate gene expression by binding to DNA motifs at enhancers and silencers, where any disruption by a SNP can potentially cause dysregulation of a target gene. Thus, we used *motifbreakR* (R package) to predict such possible events at the loci identified by our variant-to-gene mapping. Each

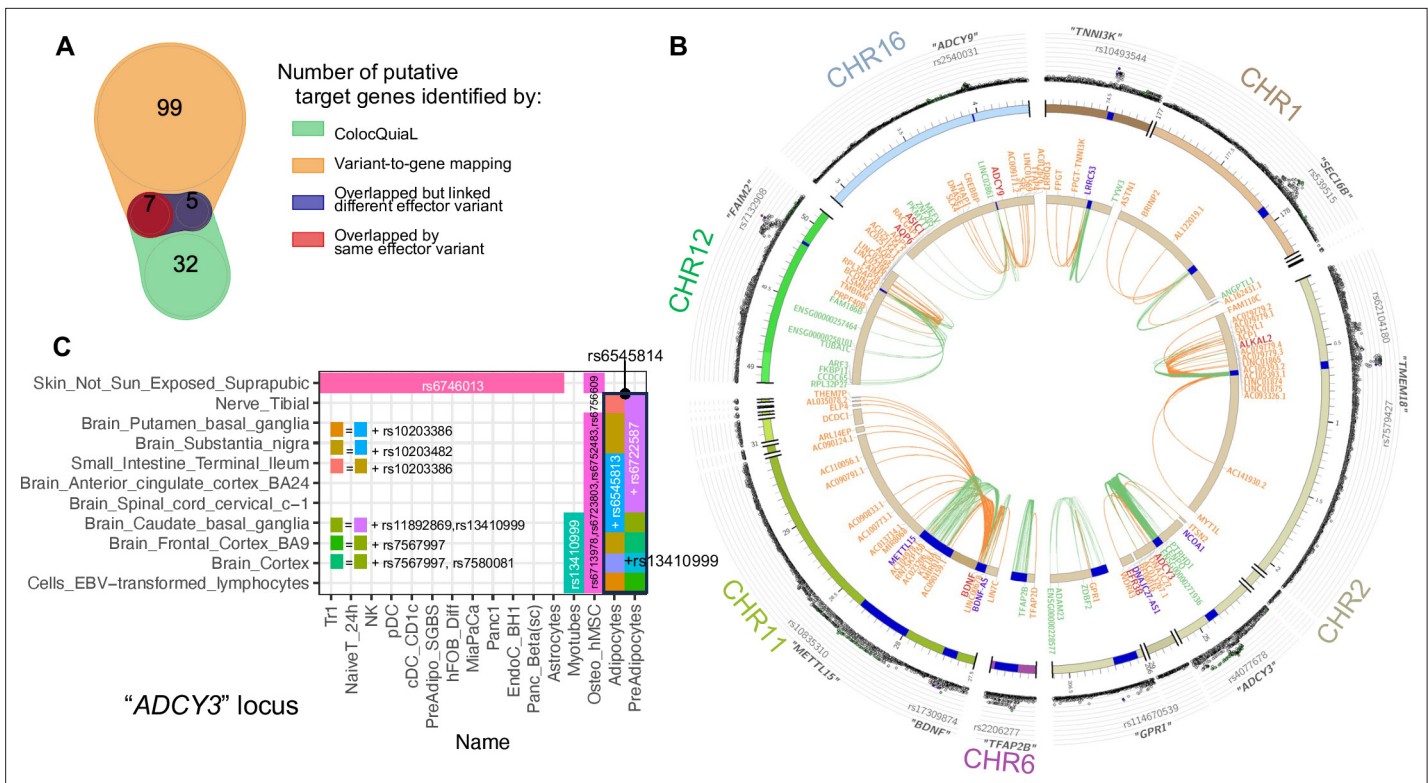

**Figure 5.** Colocalization of target effector genes with eQTLs. (**A**) Venn diagram shows the overlaps between sets of genes yielded by ColocQuiaL and the variant-to-gene mapping process. (**B**) Circos plot of the 10 loci demonstrates the differences in the ranges of associations between the two approaches, with long-ranged chromatin contacts between obesity variants and target genes displayed as orange links and short-range eQTLs colocalizations as green links. Two SNPs – rs35796073, and rs35142762 within the *TMEM18* locus, in linkage disequilibrium with rs7579427 – were estimated with high probability (cond.PP.H4=0.78) of colocalizing with the expression of *ALKAL2* gene in subcutaneous adipose tissue. These pairs of SNP-gene were also identified by our variant-to-gene mapping approach in natural killer cells, plasmacytoid dendritic cells, unstimulated PBMC naïve CD4 T cells and astrocytes. The rs7132908 variant at the *FAIM2* locus colocalized with the expression of *AQP6* in thyroid tissue and with *ASIC1* in prostate tissue, not only with high cond.PP.H4 but also with high individual SNP causal probability (SNP.PP.H4>0.95). rs7132908 was the second most consistent observation in our variant-to-gene mapping, namely across 25 different cell types (**Figure 2B**) and all three systems plus the other independent cell lines. The pair of rs7132908-contacting-*AQP6* was observed in 15 different cell types - 8 metabolic and 4 neural cell types, and 3 independent cell lines. The pair of rs7132908-contacting-*ASIC1* was observed in 11 different cell types - 8 metabolic and 2 neural cell types, and plasmacytoid dendritic cells. The other eQTL signals that overlapped with our variant-to-gene mapping results were: *BDNF* at the *METTL15* locus with its promoter physically contacted by rs11030197 in 4 cell types and its expression significantly colocalized (cond.PP.H4=0.82) in tibial artery; *ADCY9* at its locus with its promoter physically contacted by rs2531995 in natural killer cells and its expression significantly colocalized in skin tissue ('Skin_Not_Sun_Exposed_Suprapubic', cond.PP.H4=0.97). And *ADCY3* in the C panel. (**C**) ColocQuiaL estimated that these SNPs highly colocalize with the expression of *ADCY3* in 11 different tissues, where the overlapping with the 16 cell types is represented, color-coded by the proxies rs numbers.

variant was predicted to disrupt the binding of several different TFs, thus requiring further literature cross-examination to select the most probable effects. For example, rs7132908 (consistently contacting *FAIM2* in 25 cell types) was predicted to disrupt the binding of 12 different transcription factors. Among them, SREBF1 (**Figure 6A**) was the only TF that concurred with evidence that it regulates *AQP2* and *FAIM2* at the same enhancer (**Kikuchi et al., 2021**). The full prediction list can be found in **Supplementary file 1f**.

To narrow down the list of putative TF binding sites at each variant position, we leveraged the ATAC-seq footprint analysis using the RGT suite (**Li et al., 2019**). The final set of Motif-Predicted Binding Sites (MPBS) within each cell type ATAC-seq footprints was used to overlap with the genomic locations of the OCRs, and then overlapped with our obesity variants, resulting in annotated 29 variants. Mosaic plot in **Figure 6B** shows the number and proportions of variants predicted by *motifbreakR* and/or overlapped with MPBS. Insignificant p-value from Fisher's exact test indicated the independence of the two analyses. Only seven variants were found within the cREs for the same TF motifs predicted to be disrupted by *motifbreakR* (**Figure 6C**). **Figure 6—figure supplement 1**

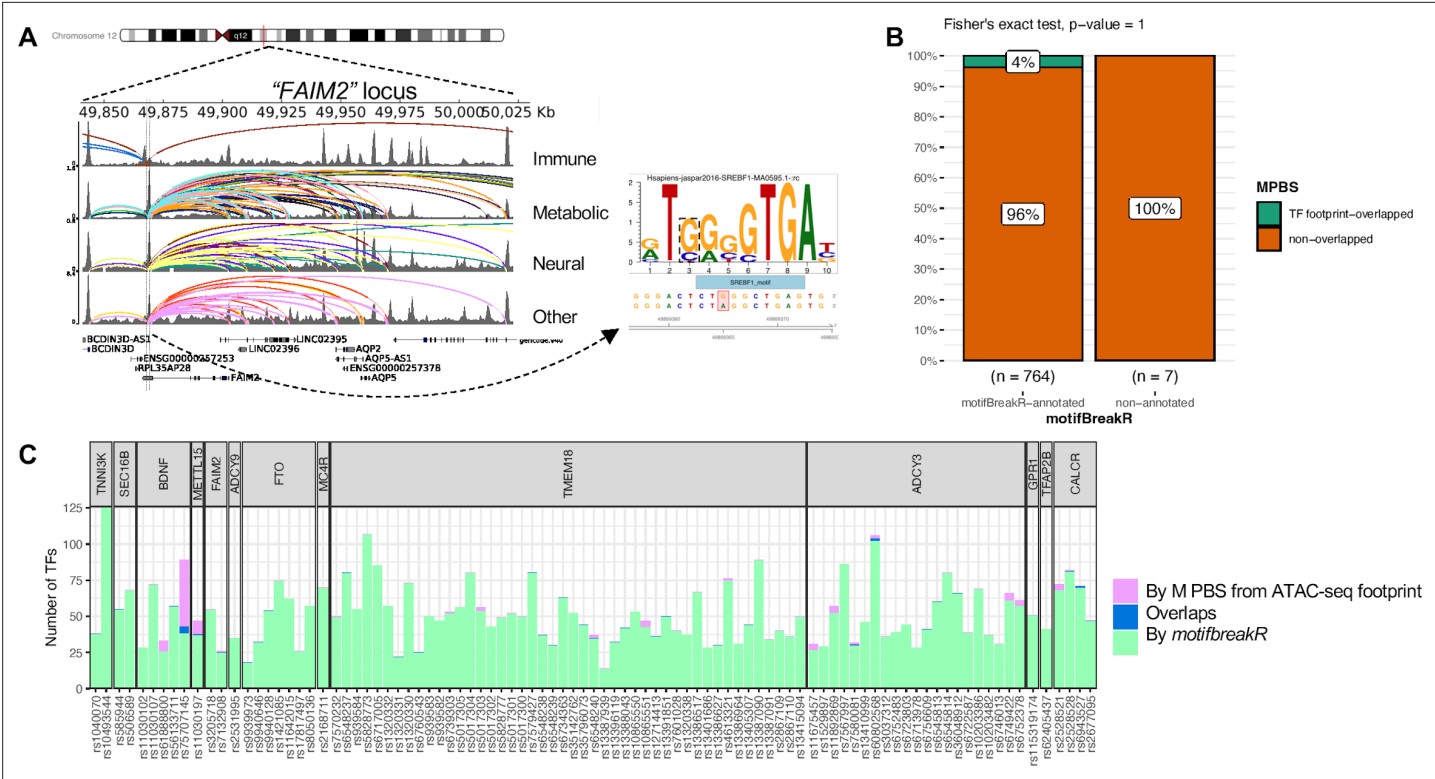

**Figure 6.** motifbreakR vs ATAC-seq footprint analysis. (**A**) Genome view at the *FAIM2* locus where rs7132908 is located and can target many genes through many chromatin contacts, presented by arcs (different colors for different cell types). rs7132908 was predicted by *motifBreakR* to disrupt the TF SREBF1's binding site, thus potentially altering the expression of its implicated genes. (**B**) Mosaic plot shows number of variants that were annotated with disrupt-TF-binding affect by motifBreakR, and the proportions that also overlapped with predicted TF footprint from ATAC-seq TF footprint analysis. Fisher exact test was performed and produced p-value = 1. (**C**) Stacked bar plot for all the variants from variant-to-gene analysis, showing number of transcription factor binding sites each of the variant can disrupt (predicted by *motifbreakR* – green), or simply overlap (analyzed by RGT suite – purple), or both (blue).

The online version of this article includes the following figure supplement(s) for figure 6:

**Figure supplement 1.** 7 variants and the corresponding TF motifs.

outlines the seven variants that *motifbreakR* and ATAC-seq footprint analysis agreed on the TF bindings they might disrupt.

## Discussion

Given the challenge of uncovering the underlying molecular mechanisms driving such a multifactorial disease as obesity, our approach leveraging GWAS summary statistics, RNA-seq, ATAC-seq, and promoter Capture C / Hi-C offers new insights. This is particularly true as it is becoming increasingly evident that multiple effector genes can operate in a temporal fashion at a given locus depending on cell state, including at the *FTO* locus (*Sobreira et al., 2021*). Our approach offers an opportunity to implicate relevant cis-regulatory regions across different cell types contributing to the genetic etiology of the disease. By assigning GWAS signals to candidate causal variants and corresponding putative effector genes via open chromatin and chromatin contact information, we enhanced the fine-mapping process with an experimental genomic perspective to yield new insights into the biological pathways influencing childhood obesity.

LD score regression is a valuable method that estimates the relationship between linkage disequilibrium score and the summary statistics of GWAS SNPs to quantify the separate contributions of polygenic effects and various confounding factors that produce SNP-based heritability of disease. The general positive heritability enrichment across our open chromatin features spanning multiple cell types (*Figure 1B*.a) reinforces the notion that obesity etiology involves many systems in our body.

While obesity has long been known to be a risk factor for pancreatitis and pancreatic cancer, the significant enrichment of pancreatic alpha and beta cell related 3D genomic features for childhood obesity GWAS signals demonstrates the bidirectional relationship between obesity and the pancreas; indeed, it is well established that insulin has obesogenic properties. Moreover, the comorbidity of obesity and diabetes (either causal or a result of the overlap between SNPs associated with these two diseases) is tangible. When focusing on genetic annotation of the cREs only, the association with obesity became more diverse across cell types, especially in metabolic cells. Interestingly, the lack of enrichment (only 8 of 57 cell types yielded no degree of enrichment) of obesity SNPs heritability in open gene promoters (*Figure 1B*.b) reveals that cRE regions harboring obesity SNPs are more involved in gene regulation than disruption, and therefore potentially contributing more weight to the manifestation of the disease.

Of course, we should factor in the effective sample sizes of the GWAS efforts that are wide-ranging (2000–24,000 – given that the N for each variant is different within a single dataset, thus contributing to the weights and p-value of each SNP when the algorithm calculates the genome-wide heritability), which could result in noise and negative enrichment observed in the analysis – a methodology limitation of partial linkage regression that has been extensively discussed in the field (*Steinsaltz et al., 2020*). Thus, it is crucial to interpret the enrichment (or lack thereof) of disease variants in a certain cellular setting with an ad hoc biological context.

From mapping the common proxies of 19 independent sentinel SNPs that were genome-wide significantly associated with childhood obesity to putative effector genes through chromatin contacting cREs, one striking finding was the several potential 'hubs' of putatively core effector genes, whose occurrence spread across three human physiological systems. With the data available from so many cell types, our approach connected new candidate causal variants to known obesity-related genes and new implications of cell modality for previously known associations.

A potential application of this association could be to fine-tune the effect of a drug toward controlling appetite. An example of bringing new aspects to the old is for the signal within the *FTO* locus that contacted *IRX3* and *IRX5*: previous studies have suggested these obesogenic effects operate in adipocytes (*Claussnitzer et al., 2016*), brain (*Smemo et al., 2014*), or pancreas *Ragvin et al., 2010*; here we confirmed this association in adipocytes and uncover the presence of distal chromatin contacts in myotubes for the first time.

Besides the above-mentioned genes with known associations with obesity, we discovered newly implicated genes. For example, the *LRRIQ3* gene at the *TNNI3K* locus had its open promoter contacted by two SNPs, rs1040070 and rs10493544, in NTERA2 cells only. The published studies (*Johnston et al., 2019*; *Sanchez-Roige et al., 2021*) that associated *LRRIQ3* with major depressive disorder and opioid usage acknowledged the overlapping promoter of this gene, albeit in the opposite direction, with a run-through transcript of *FPGT-TNNI3K* – previously shown to be associated with BMI in European *Graff et al., 2013* and Korean populations (*Lee et al., 2017*).

It is apparent that not all the implicated genes we report would contribute equally to the susceptibility of obesity pathogenesis. Each locus comprises genes whose functions are obviously related to obesity or similar traits like BMI, fat weight, etc., while other genes are not so directly obvious in their relation to these traits.

It is encouraging that for implicated genes within these multi-cell-type loci across different physiological systems we could find previous associations to the corresponding cell types or systems. Examples are the two aforementioned genes at the *TMEM18* locus (*SH3YL1* and *ACP1*) (*Fernandes et al., 2019*; *Blessing et al., 2015*; *Kobayashi et al., 2014*; *Choi et al., 2021*; *Gaynor et al., 2020*) with the broad spectrum of their functions, *HEPACAM2* implicated in the NCIH716 cell line at the *CALCR* locus *Wu et al., 2018*; *Huang et al., 2018*, and *LRRIQ3* in the NTERA2 cell line at the *TNNI3K* locus (*Pleasure and Lee, 1993*).

Chronic inflammation is an essential characteristic of obesity pathogenesis. Adipose tissue-resident immune cells have been observed, leading to an increased focus in recent years on their potential contribution to metabolic dysfunction. On the other hand, neurological or psychological conditions, such as stress, induce the secretion of both glucocorticoids (increase motivation for food) and insulin (promotes food intake and obesity). Pleasure feeding then reduces activity in the stress-response network, reinforcing the feeding habit. It has been shown that voluntary behaviors, stimulated by external or internal stressors or pleasurable feelings, memories, and habits, can override the basic

homeostatic controls of energy balance (*Dallman, 2010*). The potential link between the immune system and metabolic disease, and moreover, through the neural system, was tangible in our findings.

Two of the three SNPs which ranked the third most consistent in our variant-to-gene mapping (*Figure 2C*) – rs35796073 and rs35142762 – contacted the *ALKAL2* promoter (supported by GTEx evidence to colocalize with *ALKAL2* expression). The anaplastic lymphoma kinase (encoded by *ALK* gene) is a receptor tyrosine kinase, belongs to the insulin receptor family, and has been reported to promote nerve cell growth and differentiation (*Iwahara et al., 1997*; *Motegi et al., 2004*). Despite *ALKAL2* (ALK and LTK ligand 2) being studied principally in the context of immunity, a recent study using the EGCUT biobank GWAS identified *ALK* as a candidate thinness gene and genetic deletion showed that its expression in hypothalamic neurons acts as a negative regulator in controlling energy expenditure via sympathetic control of adipose tissue lipolysis (*Orthofer et al., 2020*). *ALKAL2* – encoding a high-affinity agonist of *ALK/LTK* receptors – which has been reported to enhance expression in response to inflammatory pain in nociceptors (*Defaye et al., 2022*; *Sun et al., 2023*) - has been recently implicated as a novel candidate gene for childhood BMI by transcriptome-wide association study *Yao et al., 2021*, and achieved genome-wide significance in a GWAS study contrasting persistent healthy thinness with severe early-onset obesity using the STILTS and SCOOP cohorts (*Riveros-McKay et al., 2019*). The finding that overexpression of *ALKAL2* could potentiate neuroblastoma progression in the absence of ALK mutation (*Borenäs et al., 2021*) echoes the relationship between *ADCY3* and *MC4R* (*Siljee et al., 2018*), where a peripheral gene, *ADCY3*, can regulate/impair the function of a core gene, that is *MC4R*, within the energy-regulating melanocortin signaling pathway (*Timshel et al., 2020*).

Our approach implicates putative target genes based on a mechanism of regulation for these variants to alter gene expression – through regulator TF(s) that bind to these contact sites. A potential limitation of the predictions from *motifbreakR* and matching TF motifs to ATAC-seq footprint by the RGT toolkit is that they were both based on the position probability matrixes of Jaspar and Hocomoco, which come from public motif databases. The ATAC-seq footprint analysis also carries sequence bias that can lead to false positive discovery. Thus, our attempt to call such regulators by predicting TF binding disruption can only serve as nominations – but warrant further functional follow up.

Another limitation of this work is the diversity in data quality among different samples, since different datasets were sampled and collected at different time points, from different patients, using different protocols, with libraries sequenced at different depths and qualities, and initially preprocessed with different pipelines and parameters. Thus, it is crucial to keep in mind that the discrepancy in data points might have resulted from variations in data quality. Importantly, any association discovered must be validated functionally before effector genes of the genetic variants can be leveraged to develop new therapies. Their putative function(s) must be characterized, together with the mechanism whereby the given variant's alleles differentially affect the expression of the targeted genes. The next step is to explore how the target genes affect the trait of interest more directly.

Our results have provided a set of leads for future exploratory experiments in specific cellular settings in order to further expand our knowledge of childhood obesity genomics and hence equip us with more effective means to overcome the burden of this systematic disease.

## Conclusion

Our approach of combining RNA-seq, ATAC-seq, and promoter Capture C/Hi-C datasets with GWAS summary statistics offers a systemic view of the multi-cellular nature of childhood obesity, shedding light on potential regulatory regions and effector genes. By leveraging physical properties, such as open chromatin status and chromatin contacts, we enhanced the fine-mapping process and gained new insights into the biological pathways influencing the disease. Although further functional validation is required, our findings provide valuable leads together with their cellular contexts for future research and the development of more effective strategies to address the burden of childhood obesity.

## Materials and methods
### Data and resource

Datasets used in prior studies are listed in *Supplementary file 1a*. ATAC-seq, RNA-seq, Hi-C, and Capture-C *library generation* for each cell type is provided in their original published study.

## Cell lines/types resources

details of each cell type/cell line source were described in their original published study, provided in *Supplementary file 1a*.

Myotubes (Primary Skeletal Muscle Cells, PCS-950–010), MiaPaCa (CRL-1420), Panc-1 (CRL-1469), microglia (HMC1, CRL-3304), human fetal osteoblastic (hFOB 1.19, CRL-3602), HepG2 hepatocarcinoma (HB-8065) and Colorectal adenocarcinoma ascites derived cells (NCIH716, CCL-251) cell lines were purchased from American Type Cell Center (ATCC, Manassas, Virginia, USA). EndoC-BH1 were purchased from Univercell Biosolutions. Primary Normal Human Astrocytes (NHA) of unknown sex were obtained from Lonza as cryopreserved cells. Human embryonic stem cells (ESC H9, WA09) were obtained from WiCell Research Institute. Melanocytes isolated from foreskin healthy newborn males, were obtained from the Specialized Programs of Research Excellence (SPORE) in Skin Cancer Specimen Resource Core at Yale University. Pancreata from deceased organ donors were obtained by Human Pancreas Analysis Program (HPAP) (https://hpap.pmacs.upenn.edu), a Human Islet Research Network consortium. All were authenticated by the source and confirmed to be mycoplasma negative.

The rest of the cell types were isolated from samples, obtained from consented human donors, maintained and expanded within corresponding laboratories. Cell identities were verified based on observed global expression variation, marker genes by PCR and immunofluorescence microscopy.

This study also includes human adipocytes and pre-adipocytes from P. Seale's lab, hepatocytes from P. Titchnell's lab, trabecular meshwork cells from J. O'Brien's lab, which will be described in detail in separate manuscripts currently in preparation.

## ATAC-seq preprocessing and peaks calling

The detailed configurations and technical details for each data set are provided in the original studies. In brief, open chromatin regions were called using ENCODE ATAC-seq pipeline as previously described in each study the published data provided. In brief, reads were aligned to the hg19 or hg38 genome using bowtie2; duplicates were removed, alignments from all replicates were pooled, and narrow peaks were called using MACS2. A region was considered open if it overlapped at least 1 bp with ATAC-seq peak. We lifted all coordinates from hg19 to hg38 to ensure consistency between datasets.

## Promoter Capture-C pre-processing and interaction calling

in brief, paired-end reads were pre-processed using HICUP pipeline (*Wingett et al., 2015*) with bowtie2 as aligner and hg19 for reference genome. Significant promoters' interactions were called using unique read pairs from all baits promoter in the reference by CHICAGO (*Cairns et al., 2016*) pipeline. In addition to analysis of individual fragments (1frag), we also binned four fragments to improve long-distance sensitivity in interactions calling (*Su et al., 2021*). Interactions with CHICAGO score >5 in either 1-fragment or 4-fragment resolution were considered significant. These interactions were output as *ibed* format (similar to BEDPE format) in which each line represents one physical contact between fragments. Interactions from both resolutions were merged and their genomic coordinates were lifted from hg19 to hg38.

## Hi-C pre-processing and interaction calling

We follow the pipeline as a recent study described (*Su et al., 2022*). Paired-end reads from each replicate were pre-processed using the HICUP pipeline v0.7.4 (*Wingett et al., 2015*), aligned by bowtie2 with hg38 as the reference genome. The alignments files were parsed to pairtools v0.3.0 to process and pairix v0.3.7 to index and compress, then converted to Hi-C matrix binary format.*cool* by cooler v0.8.11 at multiple resolutions (500 bp, 1, 2, 4, 10, 40, 500kbp and 1Mbp) and normalized with ICE method (*Imakaev et al., 2012*). The matrices from different replicates were merged at each resolution using cooler. Mustache v1.0.1 *Roayaei et al., 2020* and Fit-Hi-C2 v2.0.7 (*Kaul et al., 2020*) were used to call significant intra-chromosomal interaction loops from merged replicates matrices at three resolutions 1 kb, 2 kb, and 4 kb, with significance threshold at q-value <0.1 and FDR <1 × 10$^{-6}$, respectively. The identified interaction loops were merged between both tools at each resolution. Lastly, interaction loops from all three resolutions were merged with preference for smaller resolution if overlapped.

## Definition of cis-regulatory elements (cREs)

We intersected ATAC-seq open chromatin regions (OCRs) of each cell type with chromatin conformation capture data determined by Hi-C/Capture-C of the same cell type, and with promoters –1,500/+500 bp of TSS, which were referenced by GENCODE v30.

## Childhood obesity GWAS summary statistics

Data on childhood obesity from the EGG consortium was downloaded from https://www.egg-consortium.org/. We used 8,566,179 European ancestry variants (consisting of 8,613 cases and 12,696 controls in stage I; of 921 cases and 1930 controls in stage II), representing ~55% of the total 15,504,218 variants observed across all ancestries in the original study (*Bradfield et al., 2019*). The sumstats file was reformatted by *munge_sumstats.py* to standardize with the weighted variants from HapMap v3 within the LDSC baseline, which reduced the variants to 1,217,311 (7.8% of total).

## Cell type specific partitioned heritability

We used *LDSC* (http://www.github.com/bulik/ldsc) *v.1.0.1* (*Finucane et al., 2015*) with --h2 flag to estimate SNP-based heritability of childhood obesity within 4 defined sets of input genomic regions: (1) OCRs, (2) OCRs at gene promoters, (3) cREs, and (4) cREs with an expanded window of ± 500 bp. The baseline model LD scores, plink filesets, allele frequencies and variants weights files for the European 1000 genomes project phase 3 in hg38 were downloaded from the provided link (https://alkesgroup.broadinstitute.org/LDSCORE/GRCh38/). The cREs of each cell type were used to create the annotation, which in turn were used to compute annotation-specific LD scores for each cell types cREs set.

## Genetic loci included in variant-to-genes mapping

19 sentinel signals that achieved genome-wide significance in the trans-ancestral meta-analysis study (*Bradfield et al., 2019*) were leveraged for our analyses. Proxies for each sentinel SNP were queried using TopLD *Huang et al., 2022* and LDlinkR tool (*Myers et al., 2020*) with the GRCh38 Genome assembly, 1000 Genomes phase 3 v5 variant set, European population, and LD threshold of $r^2$ >0.8, which resulted in 771 proxies, including the 21 SNPs from the 99% credible set of the original study (*Supplementary file 1g*).

## RNA-seq preprocessing and expression profiling

The detailed configurations, steps, and technical details for each data set are provided in the original studies. In brief, read fragments from fastq files were mapped to genome assembly hg19 or hg38 using STAR, independently for each replicate and condition. We used GENCODE annotation files for feature annotation and htseq-count for raw read count calculation at each feature. Read counts were transformed into TPM (transcript per million) and normalized internally between replicates/conditions in each individual study. For comparative measurements, we transformed all the expression values into 0–100 scale.

## Differential analysis and clustering of correlated genes

Normalized transcripts per million (TPM) of all measured genes in 46 of 57 cell types was used to perform differential analysis using *DEseq2* package (*Love et al., 2014*), where cell type and system (immune, metabolic, neural and other) were used as variables for the modeling contrast. Because many of the genes we gathered from the variant-to-gene mapping were lowly expressed in corresponding cell types or others, causing relatively high levels of variability, we used *apeglm* method for effect size (logarithmic fold change estimates) shrinkage *Zhu et al., 2019* to alleviate this phenomenon during the genes ranking. Weighted correlation network analysis (WGCNA) package (*Langfelder and Horvath, 2008*) was used to cluster genes from the variant-to-genes mapping process. WGCNA network construction power was chosen based on the analysis of scale-free topology for soft-thresholding. *blockwiseModules* with a power of 10 were used to create the correlation network and cluster genes into 3 modules of similarly expressed genes (*Figure 4—figure supplement 4*). The assigned colors were used to identify the gene modules: turquoise, grey, and blue.

## Pathways enrichment analyses

We performed three analyses on the set of genes from the variant-to-genes mapping process:

*Gene set over-representation analysis (ORA)* was performed using *clusterProfiler* package **Wu et al., 2021** to identify GO biology process terms (org.Hs.eg.db databese) enriched within our genes set in each cell type. A relaxed p-value cutoff was set at 0.1, and the minimum including genes was set at 2 to ensure the capture of all possible enriched terms. An adjusted p-value of 0.05 was later used to filter the significant terms.

## Active-subnetwork-oriented gene set enrichment analysis

We used the **pathfindR** package *Ulgen et al., 2019* to identify active subnetworks in protein-protein interaction networks from Biogrid, KEGG, STRING, GeneMania, and IntAct databases, using the list of genes from the variant-to-genes mapping process. Then we provided the statistic from the differential expression analysis for **pathfindR** to perform enrichment analyses on the identified subnetworks, discovering enriched KEGG pathways.

## Customized signaling pathway impact analysis (SPIA)

The original method, proposed by Tarca in 2009 in **SPIA** package (*Tarca et al., 2009*), incorporates ORA with the adjacency matrix to measure the importance of genes within each pathway – genes that are connected to more other genes are likely more important than the downstream end-point genes. This pathway-topology approach measures the actual perturbation on a given pathway under a given condition and given differential effect size. We added more metrics to improve this method: (a) the score of gene impact among networks, (b) the neighborhood of genes that measure the importance of a gene based on its downstream effects, (c) the betweenness puts weight on the genes that act as a gateway for the network flow. The combination of metrics produces two ways of evidence – perturbance and enrichment – for each pathway similar to SPIA. Normal inverse cumulative distribution function was used to combine the p-values of this evidence, then Bonferroni and FDR correction was applied (*Figure 4—figure supplement 2*).

## GWAS-eQTL colocalization

The summary statistics for the European ancestry subset from the EGG consortium GWAS for childhood obesity was used. Common variants (MAF ≥0.01) from the 1000 Genomes Project v3 samples were used as a reference panel. We used non-overlapped genomic windows of ±250,000 bases extended in both directions from the median genomic position of each of 19 sentinel loci as input. We used ColocQuiaL *Chen et al., 2022* to test genome-wide colocalization of all possible variants included in each inputted window against GTEx v.8 eQTLs associations for all 49 tissues available from *https://www.gtexportal.org/home/datasets*. A conditional posterior probability of colocalization of 0.8 or greater was imposed.

## ATAC-seq transcription factor footprint analysis with RGT toolkit

Bam files of mapped reads from replicates and samples were merged for each cell type. The merged bam files were then used for footprinting by RGT-HINT with parameters: `--atac-seq`, `–paired-end`, *–organism = hg38*. If bam files were generated on hg19, we performed lift-over using CrossMap. py *bam* and hg19ToHg38.over.chain.gz file. We then used RGT-MOTIFANALYSIS *matching* to scan each footprint for possible transcription binding sites from HOCOMOCO and JASPAR databases for human only with parameter --filter 'species:sapiens;database:hocomoco,jaspar_vertebrates'. Parameter *–rand-proportion 10* was used to generate random putative binding sites with sizes ten times larger than the input footprints. After performing motif matching, we evaluated which transcription factors were more likely to occur in those footprints than in background regions (generated by the previous command) using RGT-MOTIFANALYSIS *enrichment* with the same filtered databases and default parameters. Output included all the Motif-Predicted Binding Sites (MPBS) that occurred within the identified footprints in each cell type. We overlapped these sites with the loci of our obesity variants.

## Prediction of variant's effect on transcription factor binding

Genomic positions (0-based coordinates) and allele alternatives of each proxy (from *SNPlocs.Hsapiens. dbSNP155.GRCh38* package with matching reference sequence from *BSgenome* package) were used

to scan all position frequency matrix databases (from *MotifDb* package) for potential transcription factor binding disruptive effects. The motifbreakR function from *motifBreakR* package was used, with *filterp = TRUE* and setting a p-value *threshold = 0.0005*, information content methods *method='ic'* with even background probabilities of the four nucleotides *bkg = c(A=0.25, C=0.25, G=0.25, T=0.25)* and *BPPARAM = BiocParallel::SerialParam()* to allow serial evaluation.

## Acknowledgements

This work was supported by National Institutes of Health awards R01 HD056465, R01 DK122586 and UM1 DK126194, and the Daniel B Burke Endowed Chair for Diabetes Research. Given the use of de-identified datasets and biospecimens was not considered human subjects research, ethical oversight was waived by the Institutional Review Board of the Children's Hospital of Philadelphia.

## Additional information

### Funding

| Funder | Grant reference number | Author |
|---|---|---|
| Eunice Kennedy Shriver National Institute of Child Health and Human Development | R01 HD056465 | Babette S Zemel<br>Alessandra Chesi<br>Andrew D Wells<br>Struan FA Grant |
| National Institute of Diabetes and Digestive and Kidney Diseases | R01 DK122586 | Andrew D Wells<br>Struan FA Grant |
| National Institute of Diabetes and Digestive and Kidney Diseases | UM1 DK126194 | Wenli Yang<br>Paul Titchenell<br>Patrick Seale<br>Klaus H Kaestner<br>Andrew D Wells<br>Struan FA Grant |

The funders had no role in study design, data collection and interpretation, or the decision to submit the work for publication.

### Author contributions

Khanh B Trang, Conceptualization, Data curation, Formal analysis, Investigation, Visualization, Methodology, Writing – original draft; Matthew C Pahl, Data curation, Formal analysis, Methodology, Writing – review and editing; James A Pippin, Joan M O'Brien, Laufey T Amundadottir, Laura Cook, Resources, Writing – review and editing; Chun Su, Data curation, Methodology; Sheridan H Littleton, Resources, Data curation, Writing – review and editing; Prabhat Sharma, Nikhil N Kulkarni, Louis R Ghanem, Natalie A Terry, Yadav Wagley, Kurt D Hankenson, Ashley Jermusyk, Jason Hoskins, Mai Xu, Kevin Brown, Stewart Anderson, Wenli Yang, Paul Titchenell, Patrick Seale, Klaus H Kaestner, Megan Levings, Resources; Babette S Zemel, Resources, Data curation, Methodology, Writing – review and editing; Alessandra Chesi, Resources, Data curation, Supervision, Methodology, Writing – review and editing; Andrew D Wells, Struan FA Grant, Conceptualization, Resources, Supervision, Funding acquisition, Writing – review and editing

### Author ORCIDs

Khanh B Trang ⓘ https://orcid.org/0000-0002-9434-508X
Louis R Ghanem ⓘ https://orcid.org/0000-0002-7723-5241
Yadav Wagley ⓘ https://orcid.org/0000-0002-1261-4267
Kurt D Hankenson ⓘ https://orcid.org/0000-0001-6361-143X
Patrick Seale ⓘ https://orcid.org/0000-0001-7119-1615
Klaus H Kaestner ⓘ https://orcid.org/0000-0002-1228-021X
Struan FA Grant ⓘ https://orcid.org/0000-0003-2025-5302

Joint public reviews: https://doi.org/10.7554/eLife.95411.3.sa1

Author response https://doi.org/10.7554/eLife.95411.3.sa2

## Additional files

### Supplementary files

Supplementary file 1. Supporting data tables. (a) Data resources of 57 cell types. (b) Variant-to-gene mapping results (c) 60 enriched KEGG pathways by pathfindR analysis (d) 39 enriched KEGG pathways by customized SPIA analysis (e) Colocalization analysis summary from ColocQuiaL. Highlighted rows are concordance with variant-to-gene mapping (f) Predicted transcription factor binding by motifbreakR to be disrupted by obesity variants (g) 771 proxies in linkage with 19 original sentinel childhood obesity signals, reported by TopLD and Ldlink.

MDAR checklist

### Data availability

The current manuscript is a computational study, so no data have been generated for this manuscript. Public software packages are available at the citations and URLs listed. Custom code for this analysis has been deposited on GitHub (copy archived at *Trang, 2024*).

The following previously published datasets were used:

| Author(s) | Year | Dataset title | Dataset URL | Database and Identifier |
|---|---|---|---|---|
| Trang KB, Levings MK, Grant SF, Wells AD | 2024 | 3D chromatin-based variant-to-gene maps across 57 human cell types reveal the cellular and genetic architecture of autoimmune disease susceptibility [captureC] | https://www.ncbi.nlm.nih.gov/geo/query/acc.cgi?&acc=GSE272080 | NCBI Gene Expression Omnibus, GSE272080 |
| Trang KB, Levings MK, Grant SF, Wells AD | 2024 | 3D chromatin-based variant-to-gene maps across 57 human cell types reveal the cellular and genetic architecture of autoimmune disease susceptibility [ATAC-seq] | https://www.ncbi.nlm.nih.gov/geo/query/acc.cgi?&acc=GSE272120 | NCBI Gene Expression Omnibus, GSE272120 |
| Trang KB, Levings MK, Grant SF, Wells AD | 2024 | 3D chromatin-based variant-to-gene maps across 57 human cell types reveal the cellular and genetic architecture of autoimmune disease susceptibility [RNA-seq] | https://www.ncbi.nlm.nih.gov/geo/query/acc.cgi?&acc=GSE271957 | NCBI Gene Expression Omnibus, GSE271957 |
| Trang KB, Levings MK, Grant SF, Wells AD | 2024 | 3D chromatin-based variant-to-gene maps across 57 human cell types reveal the cellular and genetic architecture of autoimmune disease susceptibility [HiC] | https://www.ncbi.nlm.nih.gov/geo/query/acc.cgi?&acc=GSE272172 | NCBI Gene Expression Omnibus, GSE272172 |
| Pahl MC, Sullivan KE, Romberg N, Grant SF, Wells AD | 2022 | Implicating effector genes at COVID-19 GWAS loci using promoter-focused Capture-C in disease-relevant immune cell types | https://www.ncbi.nlm.nih.gov/geo/query/acc.cgi?acc=GSE174658 | NCBI Gene Expression Omnibus, GSE174658 |
| Chesi A, Romberg N, Johnson M, Lu S, Manduchi E, Leonard M, Wells AD, Grant SF | 2020 | Promoter capture-C of primary human T Follicular Helper (TFH) cells and naive CD4-positive helper T cells from tonsils of healthy volunteers | https://www.ebi.ac.uk/biostudies/arrayexpress/studies/E-MTAB-6621 | EMBL-EBI ArrayExpress, E-MTAB-6621 |

*Continued on next page*

*Continued*

| Author(s) | Year | Dataset title | Dataset URL | Database and Identifier |
|---|---|---|---|---|
| Manduchi E, Johnson M, Chesi A, Wells AD, Grant SF, Romberg N, Lu S, Hodge K | 2020 | ATAC-seq of primary human T Follicular Helper (TFH) cells and naive CD4-positive helper T cells from tonsils of healthy volunteers | https://www.ebi.ac.uk/biostudies/arrayexpress/studies/E-MTAB-6617 | EMBL-EBI ArrayExpress, E-MTAB-6617 |
| Romberg N, Johnson M, Grant SF, Le Coz C, Manduchi E, Wells AD | 2019 | Gene expression of primary human T Follicular Helper (TFH) cells and naive CD4-positive helper T cells from tonsils of healthy volunteers | https://www.ebi.ac.uk/biostudies/arrayexpress/studies/E-MTAB-6637 | EMBL-EBI ArrayExpress, E-MTAB-6637 |
| Pahl MC, Sharma P, Wells AD | 2024 | Chromatin conformation dynamics during CD4+ T cell activation implicates 2 autoimmune disease-associated genes and regulatory elements | https://www.ncbi.nlm.nih.gov/geo/query/acc.cgi?acc=GSE230346 | NCBI Gene Expression Omnibus, GSE230346 |
| Wells AD, Su C, Manduchi E, Chesi A, Leonard M, Grant SF, Hodge K, Johnson M | 2019 | Promoter capture-C of primary HepG2 cells | https://www.ebi.ac.uk/biostudies/arrayexpress/studies/E-MTAB-7144 | EMBL-EBI ArrayExpress, E-MTAB-7144 |
| Leonard M, Grant SF, Johnson M, Wells AD, Manduchi E | 2019 | ATAC-seq of HepG2 cells | https://www.ebi.ac.uk/biostudies/arrayexpress/studies/E-MTAB-7543 | EMBL-EBI ArrayExpress, E-MTAB-7543 |
| Caliskan M, Brown CD | 2019 | Genetic and Epigenetic Fine Mapping of Complex Trait Associated Loci in the Human Liver | https://www.ncbi.nlm.nih.gov/geo/query/acc.cgi?acc=GSE128072 | NCBI Gene Expression Omnibus, GSE128072 |
| Conery M, Pippin JA, Pahl MC, Grant SF | 2024 | Bioinformatics and single-cell CRISPRi-based screen reveals effector genes and implicates multi-tissue etiology for BMD | https://www.ncbi.nlm.nih.gov/geo/query/acc.cgi?acc=GSE261284 | NCBI Gene Expression Omnibus, GSE261284 |
| Chesi A, Grant SA, Manduchi E, Wells AD, Johnson M, Hankenson K, Wagley Y | 2019 | Capture-C of primary human mesenchymal stem cells (MSC)-derived osteoblasts from healthy donors (differentiated with BPM2) | https://www.ebi.ac.uk/biostudies/arrayexpress/studies/E-MTAB-6862 | EMBL-EBI ArrayExpress, E-MTAB-6862 |
| Chesi A, Grant SF, Johnson M, Hankenson K, Wells AD, Wagley Y, Manduchi E | 2019 | ATAC-seq of primary mesenchymal stem cell (MSC)-derived osteoblasts (differentiated with BMP2) from 4 human donors | https://www.ebi.ac.uk/biostudies/arrayexpress/studies/E-MTAB-6834 | EMBL-EBI ArrayExpress, E-MTAB-6834 |
| Chesi A, Manduchi E, Grant SF, Johnson M, Hankenson K, Wells AD, Wagley Y | 2019 | RNA-seq of primary mesenchymal stem cells-derived osteoblasts (differentiated with BMP2) from 3 human donors | https://www.ebi.ac.uk/biostudies/arrayexpress/studies/E-MTAB-6835 | EMBL-EBI ArrayExpress, E-MTAB-6835 |
| Su C, Pippin J, Kaestner K, Grant SF, Wells A | 2022 | 3D chromatin organizations of human pancreatic cells reveal cell-type specific regulatory architectures of diabetes risk | https://www.ncbi.nlm.nih.gov/geo/query/acc.cgi?acc=GSE188311 | NCBI Gene Expression Omnibus, GSE188311 |
| Su C, Pippin J, Kaestner K, Grant SF, Wells A | 2022 | Single cell RNAseq and scATACseq data of Human pancreatic islets | https://hpap.pmacs.upenn.edu | Data portal of The Human Pancreas Analysis Program, PANC-DB |

*Continued on next page*

*Continued*

| Author(s) | Year | Dataset title | Dataset URL | Database and Identifier |
|---|---|---|---|---|
| Ackermann AM, Wang Z, Schug J, Naji A, Kaestner KH | 2015 | Integration of ATAC-seq and RNA-seq Identifies Human Alpha Cell and Beta Cell Signature Genes | https://www.ncbi.nlm.nih.gov/geo/query/acc.cgi?acc=GSE76268 | NCBI Gene Expression Omnibus, GSE76268 |
| Littleton SH, Grant SF, Trang KB, Chesi A | 2024 | Variant-to-function analysis of the childhood obesity chr12q13 locus implicates rs7132908 as a causal variant within the 3' UTR of FAIM2 | https://www.ncbi.nlm.nih.gov/geo/query/acc.cgi?acc=GSE241691 | NCBI Gene Expression Omnibus, GSE241691 |
| Pahl MC, Grant SF | 2021 | Cis-regulatory architecture of human ESC-derived hypothalamic neuron differentiation aids in variant-to-gene mapping of relevant common complex traits | https://www.ncbi.nlm.nih.gov/geo/query/acc.cgi?acc=GSE152098 | NCBI Gene Expression Omnibus, GSE152098 |
| Su C, Grant SF | 2022 | Promoter Capture-C of iPSC-derived NPCs and neurons | https://www.ebi.ac.uk/biostudies/arrayexpress/studies/E-MTAB-9159 | EMBL-EBI ArrayExpress, E-MTAB-9159 |
| Su C, Grant SF | 2022 | ATAC-seq of iPSC-derived NPCs and neurons | https://www.ebi.ac.uk/biostudies/arrayexpress/studies/E-MTAB-9087 | EMBL-EBI ArrayExpress, E-MTAB-9087 |
| Su C, Grant SF | 2022 | RNA-seq of iPSC-derived NPCs and neurons from CHOPWT10 and CHOPWT14 cell lines | https://www.ebi.ac.uk/biostudies/arrayexpress/studies/E-MTAB-9085 | EMBL-EBI ArrayExpress, E-MTAB-9085 |
| Johnson M, Grant SF | 2021 | Identification of 22 novel susceptibility loci associated with testicular germ cell tumors | https://www.ncbi.nlm.nih.gov/geo/query/acc.cgi?acc=GSE175368 | NCBI Gene Expression Omnibus, GSE175368 |

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
