## [Editor Report · eLife Assessment]

This **important** study presents genome-wide high-resolution chromatin-based 3D genomic interaction maps for over 50 diverse human cell types and integrates these data with pediatric obesity GWAS. The work provides **convincing** evidence that multiple pancreatic islet cell types are key effector cell types. The authors also perform variant-to-gene mapping to nominate genes underlying several GWAS hits. Overall, the results will be of interest to both the fields of 3D genome architecture and pediatric obesity.

---

## [Referee Report · Joint public reviews]

Summary:

This paper studies the genetic factors contributing to childhood obesity. Through a comprehensive analysis integrating genome-wide association study (GWAS) data with 3D genomic datasets across 57 human cell types, consisting of Capture-C/Hi-C, ATAC-seq, and RNA-seq, the study identifies significant genetic contributions to obesity using stratified LD score regression, emphasizing the enrichment of genetic signals in pancreatic alpha cells and identification of significant effector genes at obesity-associated loci such as BDNF, ADCY3, TMEM18, and FTO. Additionally, the study implicated ALKAL2, a gene responsive to inflammation in nerve nociceptors, as a novel effector gene at the TMEM18 locus, suggesting a role for inflammatory and neurological pathways in obesity's pathogenesis which was supported through colocalization analysis using eQTL derived from the GTEx dataset. This comprehensive genomic analysis sheds light on the complex genetic architecture of childhood obesity, highlighting the importance of cellular context for future research and the development of more effective strategies.

Strengths:

Overall, the paper has several strengths, including leveraging large-scale, multi-modal datasets, using appropriate computational tools, and in-depth discussion of their significant results.

---

## [Author Response]

The following is the authors’ response to the original reviews.

Response to Reviewer’s comments

We are most grateful for the opportunity to address the reviewer comments. Point-by-point responses are presented below.

Overall, the paper has several strengths, including leveraging large-scale, multi-modal datasets, using computational reasonable tools, and having an in-depth discussion of the significant results.

We thank the reviewer for the very supportive comments.

Based on the comments and questions, we have grouped the concerns and corresponding responses into three categories.

(1) The scope and data selectionThe results are somewhat inconclusive or not validated.The overall results are carefully designed, but most of the results are descriptive. While the authors are able to find additional evidence either from the literature or explain the results with their existing knowledge, none of the results have been biologically validated. Especially, the last three result sections (signaling pathways, eQTLs, and TF binding) further extended their findings, but the authors did not put the major results into any of the figures in the main text.”

The goal of this manuscript is to provide a list of putative childhood obesity target genes to yield new insights and help drive further experimentation. Moreover, the outputs from signaling pathways, eQTLs, and TF binding, although noteworthy and supportive of our method, were not particularly novel. In our manuscript we placed our focus on the novel findings from the analyses. We did, however, report the part of the eQTLs analysis concerning *ADCY3*, which brought new insight to the pathology of obesity, in Figure 4C.

The manuscript would benefit from an explanation regarding the rationale behind the selection of the 57 human cell types analyzed. it is essential to clarify whether these cell types have unique functions or relevance to childhood development and obesity.

We elected to comprehensively investigate the GWAS-informed cellular underpinnings of childhood development and obesity. By including a diverse range of cell types from different tissues and organs, we sought to capture the multifaceted nature of cellular contributions to obesity-related mechanisms, and open new avenues for targeted therapeutic interventions.

There are clearly cell types that are already established as being key to the pathogenesis of obesity when dysregulated: adipocytes for energy storage, immune cell types regulating inflammation and metabolic homeostasis, hepatocytes regulating lipid metabolism, pancreatic cell types intricately involved in glucose and lipid metabolism, skeletal muscle for glucose uptake and metabolism, and brain cell types in the regulation of appetite, energy expenditure, and metabolic homeostasis.

While it is practical to focus on cell types already proven to be associated with or relevant to obesity, this approach has its limitations. It confines our understanding to established knowledge and rules out the potential for discovering novel insights from new cellular mechanisms or pathways that could play significant roles in the pathogenesis if obesity. Therefore, it was essential to reflect known biology against the unexplored cell types to expand our overall understanding and potentially identify innovative targets for treatment or prevention.

I wonder whether the used epigenome datasets are all from children. Although the authors use literature to support that body weight and obesity remain stable from infancy to adulthood, it remains uncertain whether epigenomic data from other life stages might overlook significant genetic variants that uniquely contribute to childhood obesity.

The datasets utilized in our study were derived from a combination of sources, both pediatric and adult. We recognize that epigenetic profiles can vary across different life stages but our principal effort was to characterize susceptibility BEFORE disease onset.

Given that the GTEx tissue samples are derived from adult donors, there appears to be a mismatch with the study's focus on childhood obesity. If possible, identifying alternative validation strategies or datasets more closely related to the pediatric population could strengthen the study's findings.

We thank the reviewer for raising this important point. We acknowledge that the GTEx tissue samples are derived from adult donors, which might not perfectly align with the study's focus on childhood obesity. The ideal strategy would be a longitudinal design that follows individuals from childhood into adulthood to bridge the gap between pediatric and adult data, offering systematic insights into how early-life epigenetic markers influencing obesity later in life. In future work, we aim to carry out such efforts, which will represent substantial time and financial commitment.

Along the same lines, the Developmental Genotype-Tissue Expression (dGTEx) Project is a new effort to study development-specific genetic effects on gene expression at 4 developmental windows spanning from infant to post-puberty (0-18 years). Donor recruitment began in August 2023 and remains ongoing. Tissue characterization and data production are underway. We hope that with the establishment of this resource, our future research in the field of pediatric health will be further enhanced.

Figure 1B: in subplots c and d, the results are either from Hi-C or capture-C. Although the authors use different colors to denote them, I cannot help wondering how much difference between Hi-C and capture-C brings in. Did the authors explore the difference between the Hi-C and capture-C?

Thank you for your comment. It is not within the scope of our paper to explore the differences between the Hi-C and Capture-C methods. In the context of our study, both methods serve the same purpose of detecting chromatin loops that bring putative enhancers to sometimes genomically distant gene promoters. Consequently, our focus was on utilizing these methods to identify relevant chromatin interactions rather than comparing their technical differences.

(2) Details on defining different categories of the regions of interestSome technical details are missing.While the authors described all of their analysis steps, a lot of the time, they did not mention the motivation. Sometimes, the details were also omitted.”

We have added a section to the revision to address the rationale behind different OCRs categories.

Line 129: should "-1,500/+500bp" be "-500/+500bp"?

A gene promoter was defined as a region 1,500 bases upstream to 500 bases downstream of the TSS. Most transcription factor binding sites are distributes upstream (5’) from TSS, and the assembly of transcription machinery occurs up to 1000 bases 5’ from TSS. Given our interest in SNPs that can potentially disrupt transcription factor binding, this defined promoter length allowed us to capture such SNPs in our analyses.

How did the authors define a contact region?

Chromatin contact regions identified by Hi-C or Capture-C assays are always reported as pairs of chromatin regions. The Supplementary eMethods provide details on the method of processing and interaction calling from the Hi-C and Capture-C data.

The manuscript would benefit from a detailed explanation of the methods used to define cREs, particularly the process of intersecting OCRs with chromatin conformation data. The current description does not fully clarify how the cREs are defined.In the result section titled "Consistency and diversity of childhood obesity proxy variants mapped to cREs", the authors introduced the different types of cREs in the context of open chromatin regions and chromatin contact regions, and TSS. Figure 2A is helpful in some way, but more explanation is definitely needed. For example, it seems that the authors introduced three chromatin contacts on purpose, but I did not quite get the overall motivation.

We apologize for the confusion. Our definition of cREs is consistent throughout the study. Figure 2A will be the first Figure 1A in the revision in order to aid the reader.

The 3 representative chromatin loops illustrate different ways the chromatin contact regions (pairs of blue regions under blue arcs) can overlap with OCRs (yellow regions under yellow triangles – ATAC peaks) and gene promoters.

(1) The first chromatin loop has one contact region that overlaps with OCRs at one end and with the gene promoter at the other. This satisfies the formation of cREs; thus, the area under the yellow ATAC-peak triangle is green.

(2) The second loop only overlapped with OCR at one end, and there was no gene promoter nearby, so it is unqualified as cREs formation.

(3) The third chromatin loop has OCR and promoter overlapping at one end. We defined this as a special cRE formation; thus, the area under the yellow ATAC-peak triangle is green.

To avoid further confusion for the reader, we have eliminated this variation in the new illustration for the revised manuscript.

Figure 2A: The authors used triangles filled differently to denote different types of cREs but I wonder what the height of the triangles implies. Please specify.

The triangles are illustrations for ATAC-seq peaks, and the yellow chromatin regions under them are OCRs. The different heights of ATAC-seq peaks are usually quantified as intensity values for OCRs. However, in our study, when an ATAC-seq peak passed the significance threshold from the data pipeline, we only considered their locations, regardless of their intensities. To avoid further confusion for the reader, we have eliminated this variation in the new illustration for the revised manuscript.

Figure 1B-c. the title should be "OCRs at putative cREs". Similarly in Figure 1B-d.

cREs are a subset of OCRs.

- In the section "Cell type specific partitioned heritability", the authors used "4 defined sets of input genomic regions". Are you corresponding to the four types of regions in Figure 2A?

Figure 2A is the first Figure 1A in the revision and is modified to showcase how we define OCRs and cREs.

It seems that the authors described the 771 proxies in "Genetic loci included in variant-to-genes mapping" (ln 154), and then somehow narrowed down from 771 to 94 (according to ln 199) because they are cREs. It would be great if the authors could describe the selection procedure together, rather than isolated, which made it quite difficult to understand.

In the Methods section entitled “Genetic loci included in variant-to-genes mapping," we described the process of LD expansion to include 771 proxies from 19 sentinel obesity-significantly associated signals. Not all of these proxies are located within our defined cREs. Figure 2B, now Figure 2A in the revision, illustrates different proportions of these proxies located within different types of regions, reducing the proxy list to 94 located within our defined cREs.

Figure 2. What's the difference between the 771 and 758 proxies?

13 out of 771 proxies did not fall within any defined regions. The remaining 758 were located within contact regions of at least one cell type regardless of chromatin state.

(3) TyposIn the paragraph "Childhood obesity GWAS summary statistics", the authors may want to describe the case/control numbers in two stages differently. "in stage 1" and "921 cases" together made me think "1,921" is one number.

This has been amended in the revision.

Hi-C technology should be spelled as Hi-C. There are many places, it is miss-spelled as "hi-C". In Figure 1, the author used "hiC" in the legend. Similarly, Capture-C sometime was spelled as "capture-C" in the manuscript.At the end of the fifth row in the second paragraph of the Introduction section: "exisit" should be "exist".In Figure 2A: "Within open chromatin contract region" should be "Within open chromatin contact region”

These typos and terminology inconsistencies have been amended in the revision.